# Neutralizing human monoclonal antibodies that target the PcrV component of the type III secretion system of *Pseudomonas aeruginosa* act through distinct mechanisms

**Jean-Mathieu Desveaux**[1†], **Eric Faudry**[1†], **Carlos Contreras-Martel**[1],
**François Cretin**[1], **Leonardo Sebastian Dergan-Dylon**[1], **Axelle Amen**[1,2],
**Isabelle Bally**[1], **Victor Tardivy-Casemajor**[1], **Fabien Chenavier**[1],
**Delphine Fouquenet**[3], **Yvan Caspar**[1,4], **Ina Attree**[1*], **Andrea Dessen**[1*],
**Pascal Poignard**[1,5*]

[1]Univ. Grenoble Alpes, CEA, CNRS, Institut de Biologie Structurale (IBS), Grenoble,
France; [2]Laboratoire d'Immunologie, CHU Grenoble Alpes, Grenoble, France; [3]Centre
d'Étude des Pathologies Respiratoires INSERM U1100 - UFR de Médecine de Tours,
Tours, France; [4]Laboratoire de Bactériologie-Hygiène Hospitalière, CHU Grenoble
Alpes, Grenoble, France; [5]Laboratoire de Virologie, CHU Grenoble Alpes, Grenoble,
France

**\*For correspondence:**
ina.attree@ibs.fr (IA);
andrea.dessen@cnrs.fr (AD);
pascal.poignard@ibs.fr (PP)

[†]These authors contributed
equally to this work

**Competing interest:** The authors
declare that no competing
interests exist.

**Reviewing Editor:** Alan Talevi,
Universidad Nacional de La Plata,
Argentina

## eLife Assessment

This **useful** work identifies new monoclonal antibodies produced by cystic fibrosis patients against *Pseudomonas aeruginosa* type three secretion system. The evidence supporting authors' claim is **solid**. Nonetheless, the manuscript may benefit from a more in depth description of what the authors learned from their structure-based analyses of antibodies targeting PcrV.

**Abstract** *Pseudomonas aeruginosa* is a major human opportunistic pathogen associated with a high incidence of multi-drug resistance. The antibody-based blockade of *P. aeruginosa* virulence factors represents a promising alternative strategy to mitigate its infectivity. In this study, we employed single B cell sorting from cystic fibrosis patients to isolate human monoclonal antibodies (mAbs) targeting proteins from the *P. aeruginosa* Type 3 Secretion System (T3SS) and characterized a panel of mAbs directed at PscF and PcrV. Among those, two mAbs, P5B3 and P3D6, that bind to the injectisome tip protein PcrV, exhibited T3SS blocking activity. We solved the crystal structure of the P3D6 Fab-PcrV complex, which revealed that the Ab binds to the C-terminal region of PcrV. In addition, we compared the T3SS-blocking activity of three PcrV-targeting mAbs, including two from previous independent studies, using two distinct assays to evaluate pore formation and toxin injection. We conducted a mechanistic and structural analysis of their modes of action through modeling based on the known structure of a functional homolog, SipD from *Salmonella typhimurium*. The analysis suggests that anti-PcrV mAbs may act through different mechanisms, ranging from preventing PcrV oligomerization to disrupting PcrV's scaffolding function, thereby inhibiting the assembly and function of the translocon pore. Our findings provide additional evidence that T3SS-targeting Abs, some capable of inhibiting virulence, are elicited in *P. aeruginosa*-infected patients. The results offer deeper insights into PcrV recognition by mAbs and their associated mechanisms of action, helping

to identify which Abs are more likely to be therapeutically useful based on their mode of action and potency. This paves the way for the development of effective alternatives to traditional antibiotics in the fight against this resilient pathogen.

## Introduction

The emergence of antimicrobial resistance is a major threat to human health. Among the micro-organisms whose resistance rates have increased the most dramatically are ESKAPE pathogens (*Enterococcus faecium, Staphylococcus aureus, Klebsiella pneumoniae, Acinetobacter baumannii, Pseudomonas aeruginosa, Enterobacter spp.*) for which novel antibacterial treatments are urgently needed. However, an antibiotic discovery hiatus that occurred during the last few decades severely heightened the resistance threat (*Murray et al., 2022*), underlining the importance of exploring alternative strategies, such as host-targeting, bacteriophage, anti-virulence, and Ab-based therapies (*de Melo et al., 2024*; *Kaufmann et al., 2018*; *Morrison, 2015*).

Therapeutic mAbs have been successfully developed to fight viral infections ranging from Ebola to SARS-CoV-2 (*Crowe, 2022*; *Levin et al., 2022*; *Mulangu et al., 2019*). To date, however, only three therapeutic Abs have been marketed against bacteria, all of which target toxins. Other types of bacterial virulence factors could also serve as potential high importance targets for mAbs. Recent examples include the development of mAbs that target lipopolysaccharides, O-antigen, and outer membrane transporter proteins, notably in *Klebsiella pneumoniae* and *Mycobacterium tuberculosis* (*Pennini et al., 2017*; *Rollenske et al., 2018*; *Watson et al., 2021*). The advantages of targeting virulence factors through mAbs include notably a high specificity and the decreased likelihood of the emergence of resistance among bacteria (*La Guidara et al., 2024*). Additionally, the employment of mAb engineering platforms offers the potential for improved efficacy through modifications, such as half-life extension and alterations of Fc effector functions (*Morrison, 2015*; *Vacca et al., 2022*). Finally, strategies such as the use of mAb cocktails targeting different specificities and the combination with traditional antibiotics further expand the range of Ab-based treatment options (*Duan et al., 2021*; *Morrison, 2015*; *Tabor et al., 2018*).

*Pseudomonas aeruginosa* is a major nosocomial pathogen and the leading cause of acute pneumonia and chronic lung infections, particularly in ventilator-assisted and cystic fibrosis (CF) patients. Infections with *P. aeruginosa* ultimately lead to loss of lung function and death in CF patients. Worldwide, *P. aeruginosa* is responsible for more than 300,000 deaths associated or attributed to resistance each year. The natural resistance of *P. aeruginosa* to a broad range of antibiotics, its ability to grow as biofilms, as well as its widespread presence in hospital settings (*Horcajada et al., 2019*; *Murray et al., 2022*), have called for urgent efforts towards the development of new therapeutic agents. Aggressive acute infections by *P. aeruginosa* are highly dependent on its T3SS, a needle-like, multicomponent secretion machinery located on the cell surface and that transports effectors from the bacterial cytoplasm directly into the host cell cytosol (*Goure et al., 2004*; *Hauser, 2009*; *Quinaud et al., 2007*; *Quinaud et al., 2005*). It is of note that in other human pathogens, such as *Yersinia pestis*, *Salmonella typhi*, *Shigella dysenteriae,* and *Escherichia coli,* the T3SS also plays a key role in virulence, participating in the causation of diseases, such as plague, typhoid fever, and bacillary dysentery, respectively (*Coburn et al., 2007*; *Diepold and Wagner, 2014*; *Hu et al., 2017*; *Serapio-Palacios and Finlay, 2020*).

A key component of the T3SS is the injectisome, membrane-embedded protein rings extended by a hollow needle, composed of the PscF protein that protrudes outwards from the bacterial surface. Injectisome-dependent toxin delivery, which occurs upon contact with the eukaryotic target cell, also requires formation of the 'translocon,' a complex of three proteins that are exported through the interior of the polymerized needle, assemble at its tip, and form a pore in the eukaryotic cell membrane, an essential step for effector injection (*Mueller et al., 2008*). The translocon is composed of two hydrophobic proteins (PopB and PopD in *P. aeruginosa*), as well as a hydrophilic partner–PcrV, or the V antigen–in *P. aeruginosa* (*Matteï et al., 2011*). PopB and PopD have been shown to act as *bona fide* pore-forming toxins (*Schoehn et al., 2003*; *Faudry et al., 2006*; *Montagner et al., 2011*) that, upon membrane disruption, can trigger the manipulation of host processes, including histone dephosphorylation and mitochondrial network disruption (*Dortet et al., 2018*). PcrV, on the other hand, oligomerizes at the tip of the T3SS needle and aids PopB and PopD in their membrane disruption process

(*Gébus et al., 2009*; *Goure et al., 2005*; *Guo and Galán, 2021*; *Matteï et al., 2011*). Crystal structures of monomeric homologs of PcrV (LcrV, SipD, BipD) have shown that they fold into an elongated coiled-coil buttressed by an a-helical hairpin at the N-terminus and an α/β carboxy-terminal region (*Derewenda, 2011*; *Erskine et al., 2006*; *Lunelli et al., 2011*). Notably, the cryo-EM structure of a needle filament complex composed of PrgI (needle protein) with SipD (tip protein) at its extremity confirmed that the latter forms a pentamer where the first and fifth subunits are separated by a gap, thus generating a heterogeneous assembly (*Guo and Galán, 2021*). This arrangement could be similar in numerous T3SS systems (*Habenstein et al., 2019*).

Given the importance of the T3SS for *P. aeruginosa* infection, components such as PcrV and PscF have been explored as targets for the development of therapeutic Abs and inhibitory small molecules, respectively (*Berube et al., 2017*; *Bowlin et al., 2014*). Animal models have shown that blocking the T3SS, particularly the function of PcrV, can successfully diminish tissue damage due to *P. aeruginosa* infection (*Frank et al., 2002*; *Imamura et al., 2007*). Moreover, in ventilated patients, pegylated Fabs that target PcrV (KB001-A) were shown to successfully reduce the incidence of pneumonia, which is consistent with the role of T3SS in the acute phase of infection (*Jain et al., 2018*; *Roy-Burman et al., 2001*). However, this treatment did not benefit chronically colonized CF patients in terms of antibiotic needs (*François et al., 2012*; *Jain et al., 2018*; *Yaeger et al., 2021*). In addition, the bispecific MEDI3902 mAb targeting both PcrV and the Psl exopolysaccharide successfully protected against *P. aeruginosa* infection in animal models but was discontinued in phase II clinical trials (*Chastre et al., 2022*; *DiGiandomenico et al., 2014*). Nevertheless, PcrV remains an attractive target, motivating the search for potentially more effective Abs (*Simonis et al., 2023*).

Here, we sorted specific single memory B cells from peripheral blood mononuclear cells (PBMCs) of cystic fibrosis patients to identify mAbs against PcrV and PscF with potential T3SS-inhibiting activity. Two anti-PcrV mAbs (P5B3 and P3D6) showed inhibition of the injection of the T3SS effector ExoS into epithelial cells, with mAb P5B3 displaying blocking activity against five major PcrV variants representing more than 80% of clinical isolates sequenced to date. We obtained the crystal structure of a P3D6 Fab-PcrV complex and further compared the mechanisms of action of different anti-PcrV mAbs targeting various epitopes, including one mAb from a recent publication (*Simonis et al., 2023*). These structure-based analyses of the mechanisms of action of the different mAbs provide valuable insights for the development of improved antipseudomonal treatments and preventive approaches.

## Results

### Selection of donors exhibiting T3SS-inhibiting circulating IgG responses

Our approach was based on single cell sorting of recombinant PcrV and PscF-specific memory B cells from human donor PBMCs. To identify donors with anti-PcrV and -PscF mAbs with T3SS-inhibitory activity, we first evaluated in ELISA the reactivity of sera from a cohort of CF patients that were chronically colonized with *P. aeruginosa* against recombinant PcrV and PscF before testing them in functional assays (*Figure 1A*). Among the 34 sera tested, donors 16, and 25 exhibited the strongest reactivity for both proteins (*Figure 1B*).

To assess the capacity of Protein A-purified serum IgGs (predominantly IgG1, IgG2, and IgG4) to block T3SS effector translocation, we used a previously developed cellular model that is based on the T3SS-dependent translocation of the ExoS effector fused to β-lactamase, ExoS-Bla (*Verove et al., 2012*). Briefly, epithelial cells were exposed to *P. aeruginosa* CHAΔ*exoS* expressing the ExoS-Bla reporter in the presence of patients' polyclonal purified IgGs. ExoS-Bla translocation was measured by monitoring fluorescence of the β-lactamase FRET-competent substrate CCF2-AM, and expressed as normalized reporter injection. Polyclonal IgGs from donors 16 and 25 showed a potent ExoS-Bla translocation blocking activity with an almost complete inhibition of injection at 160 µg/mL (*Figure 1C*). To investigate whether the observed activity was driven by anti-PcrV and/or anti-PscF specific IgGs, we absorbed specific Abs on beads coated with recombinant PcrV or PscF to obtain polyclonal IgG samples depleted of the corresponding specific IgGs (*Figure 1D*, top). The T3SS-blocking activity of depleted polyclonal IgGs was then evaluated using the same method as above (*Figure 1D*, bottom). The results showed a decrease in inhibitory activity when anti-PcrV Abs were depleted from donor 25's IgGs and when anti-PscF Abs were depleted from donor 16's IgGs, suggesting the presence of inhibitory Abs against the respective proteins. Additionally, the findings demonstrated that our

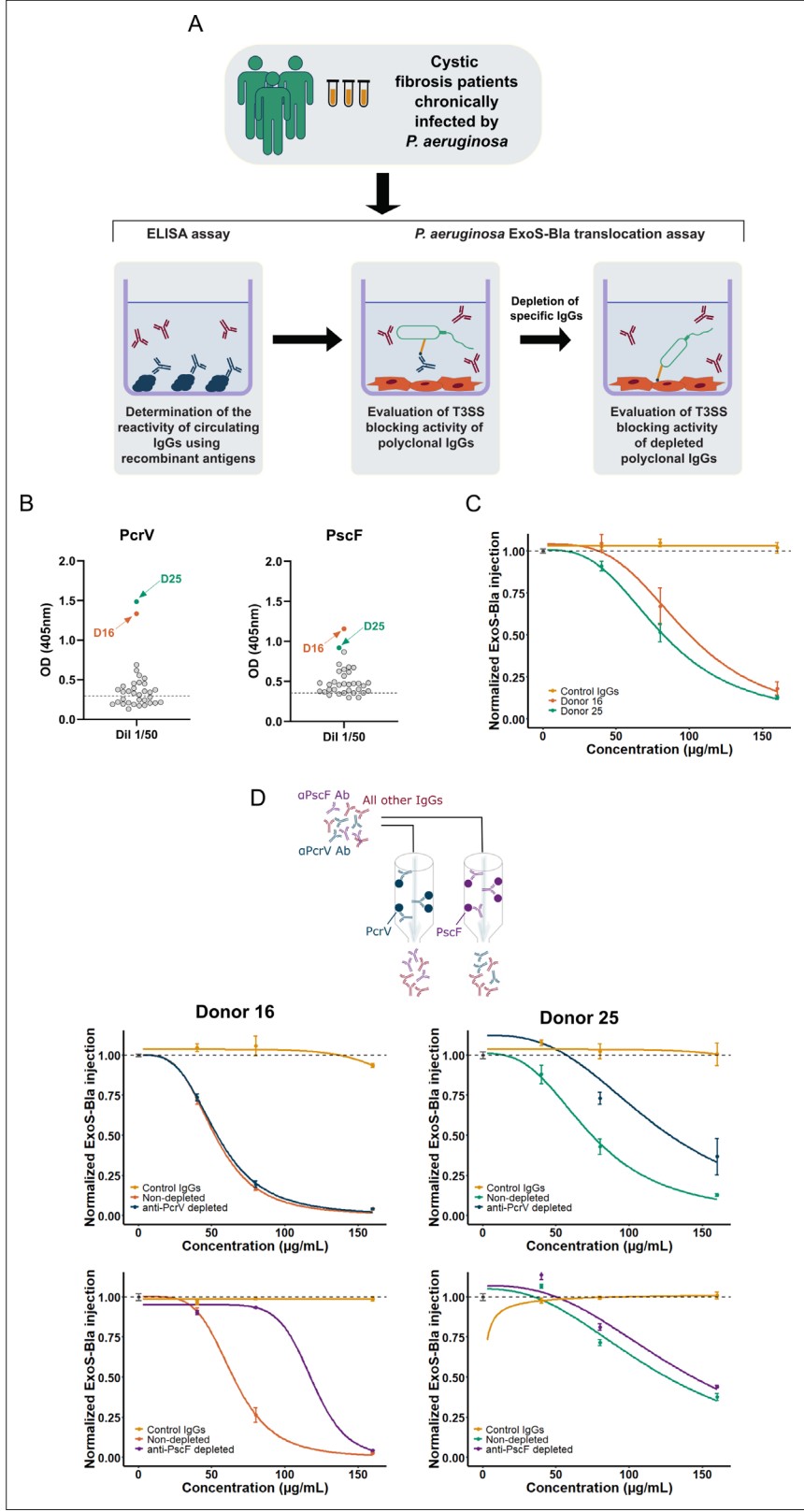

**Figure 1.** Screening workflow and donor selection. (**A**) Schematic representation of the workflow from patient selection to evaluation of Type 3 Secretion System (T3SS)-blocking activity. (**B**) Patients' sera (1/50 dilution) were tested in ELISA against recombinant PcrV and PscF. (**C**) ExoS-Bla translocation blocking activity of serum IgGs from donors 16 and 25. The dots and bars represent the means and standard deviations of data from three (Donor 16)

*Figure 1 continued on next page*

*Figure 1 continued*

and two (Donor 25) experiments with three technical replicates each. (**D**) (top) scheme of depletion experiment of specific Abs on either PscF- or PcrV-loaded columns. (bottom) blocking activity of depleted sera for both donors. The dots and bars represent the means and standard deviations of experimental triplicates. The curves correspond to the modeled log-logistic dose-response curves. The dashed lines represent the mean of normalized ExoS-Bla injection in the absence of Ab. Source Data: *Figure 1—source data 1*.

The online version of this article includes the following source data for figure 1:

**Source data 1.** Raw data for plots of *Figure 1* - Donor selection.

recombinant antigen baits could effectively bind T3SS-inhibitory Abs and could, therefore, be used to isolate memory B cells producing the corresponding IgGs.

## Isolation of PcrV and PscF mAbs using a single-cell direct sorting approach

To isolate mAbs specific to PcrV and PscF, PBMCs were purified from whole blood from the two selected donors. Next, using single-cell sorting, IgG-positive memory B cells were isolated based on their ability to recognize either PscF or PcrV (*Figure 2A*) and seeded at the frequency of one cell per well. Variable heavy and light chain gene sequences were retrieved from isolated B cells leading to the production of a total of 66 recombinant mAbs (53 and 13 putative anti-PscF and anti-PcrV, respectively). The specific binding capacities of 10 anti-PscF and four anti-PcrV mAbs were confirmed by ELISA against the corresponding recombinant proteins. $EC_{50}$ values calculated from ELISA data showed variable apparent affinities ranging from ~50 µg/mL to 0.02 µg/mL (*Figure 2B*). Isolated mAbs originated from a variety of variable gene germline families, as determined using the international immunogenetics information system (IMGT) database alignments, and did not present any notable features in terms of mutation rates or HCDR3 length (*Supplementary file 1*), with no particular enrichment noted.

The ability of ELISA-confirmed anti-PscF and anti-PcrV mAbs to block T3SS-mediated activity at a concentration of 100 µg/mL was subsequently evaluated using the ExoS-Bla reporter system. No significant reduction in ExoS-Bla injection was observed for any of the anti-PscF mAbs tested. However, two out of four anti-PcrV mAbs, P5B3, and P3D6, significantly reduced ExoS-Bla injection, with P3D6 displaying significantly stronger efficacy (*Figure 2B*).

To map the epitopes of the isolated mAbs, we next performed competition ELISAs (*Supplementary file 2*). Antibodies directed at PscF grouped into three clusters, with P1D8 and P5G10 mAbs competing only against themselves. Similarly, anti-PcrV mAbs also grouped into three clusters, with the two anti-PcrV mAbs exhibiting T3SS inhibitory activity, P5B3, and P3D6, seemingly targeting overlapping epitopes. Precise affinities of both mAbs were measured using biolayer interferometry (BLI), revealing sub-nanomolar $K_D$ values (*Supplementary file 3*). Notably, P3D6 exhibited approximately 30-fold lower affinity compared to P5B3, despite demonstrating greater efficacy in the inhibitory assay.

## P5B3 inhibits T3SS-dependent toxin injection by recognizing a highly conserved epitope of PcrV

Polymorphism in PcrV protein sequences was reported among *P. aeruginosa* clinical isolates and should be considered in the development of therapeutic human monoclonal Abs targeting PcrV (*Figure 3A*; *Tabor et al., 2018*). To determine whether the blocking activity of mAbs P5B3 and P3D6 was impacted by the PcrV sequence, the reporter ExoS-Bla was introduced into a strain that lacked isogenic PcrV (Δ*pcrV*) and synthesized the five most prevalent PcrV variants found in over 80% of clinical isolates (*Tabor et al., 2018*). Monoclonal Ab P5B3 showed statistically significant T3SS blocking activity towards all variants (*Figure 3B*) with estimated $IC_{50}$ values ranging from 100 µg/mL to 400 µg/mL for the five variants (no statistically significant difference; *Figure 3—figure supplement 1*). In contrast, mAb P3D6 had no effect on variants 2, 3, 4, and 5, but strongly inhibited variant 1 (*Figure 3B*) with an estimated $IC_{50}$ of 3.7 µg/mL (*Figure 3—figure supplement 1*), indicating that the epitope recognized by P3D6 differs between PcrV variants.

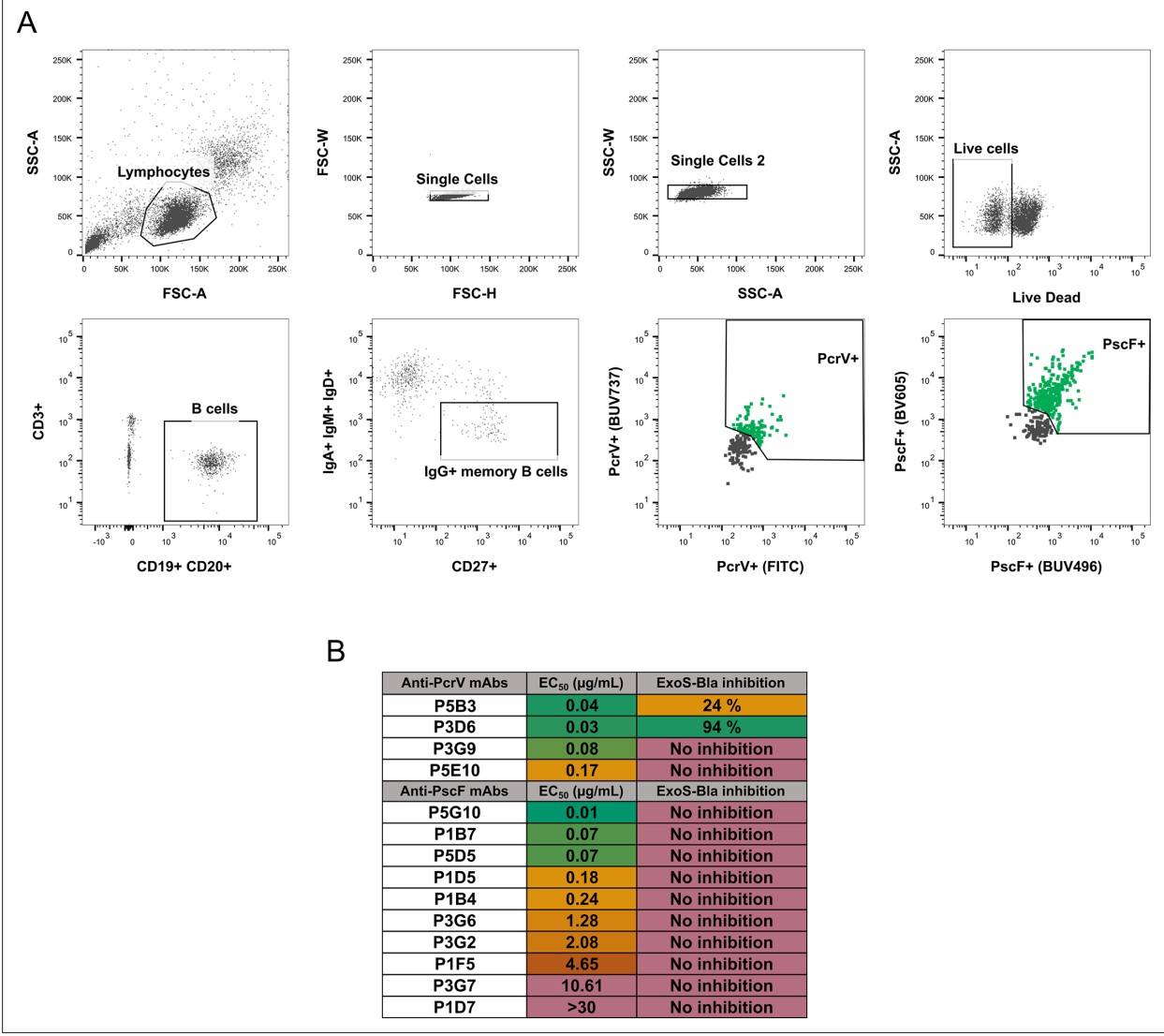

**Figure 2.** Selection of B cells from donors 16 and 25. (**A**) B cell sorting and isolation using PscF and PcrV baits. (**B**) Table summarizing the EC$_{50}$ values of selected Abs obtained by ELISA and the percentage of inhibition of ExoS-Bla injection into epithelial cells at 100 µg/mL. ExoS-Bla inhibitions were compared using ANOVA and 'No inhibition' means an absence of significant difference with the control (adjusted $p$-values >0.05). The P5B3 and P3D6 mAbs exhibited differences with the control (no Ab) with adjusted $p$-values <0.001. Source Data: *Figure 2—source data 1*.

The online version of this article includes the following source data for figure 2:

**Source data 1.** Raw data for table of *Figure 2* - Antibody affinity and inhibitory activity.

## Anti-PcrV mAbs block translocon pore assembly

It has been suggested that PcrV scaffolds the assembly of the PopB/PopD translocon within host membranes by interacting with the PopD component of the pore (*Goure et al., 2004*; *Kundracik et al., 2022*; *Kundracik et al., 2022*; *Matteï et al., 2011*). Furthermore, polyclonal Abs raised against PcrV have been shown to inhibit the assembly of the translocon in target membranes (*Goure et al., 2005*).

To investigate the mechanistic details of the inhibitory activity of mAbs P3D6 and P5B3, we used a *P. aeruginosa* strain deprived of all three T3SS effectors, ExoS, ExoT, and ExoY. This strain, named PAO1Δ3Tox (*Cisz et al., 2008*), harbors PcrV variant 1 and provokes toxin-independent macrophage pyroptosis upon membrane insertion of the PopB/PopD translocation pore (*Dacheux et al., 2001*). Death of J774 macrophages was monitored during 4 hr post-infection by measuring an increase in propidium iodide fluorescence due to DNA binding to the nuclei of dead cells. Both mAbs significantly

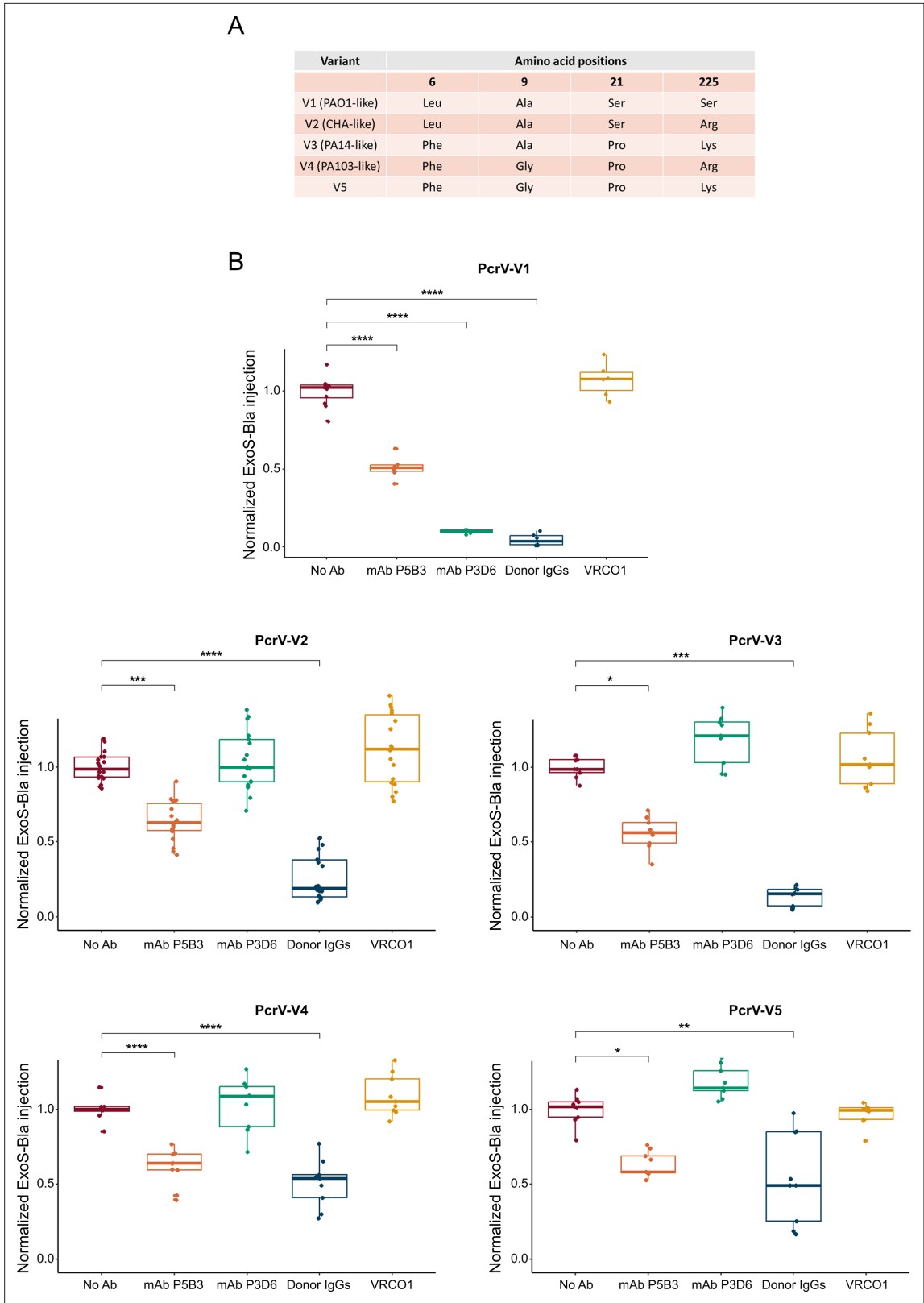

**Figure 3.** Monoclonal antibodies (mAbs) P5B3 and P3D6 activity on PcrV variants. (**A**) PcrV variability in clinical strains. The most variable position (225) can either be Ser, Arg, or Lys. Representative strains are indicated when available (PAO1 for V1, CHA for V2, PA14 for V3, and PA103 for V4). (**B**) Inhibition of ExoS-Bla activity following infection of A549 epithelial cells with *P. aeruginosa* expressing the PcrV variants. Normalized ExoS-Bla injection values in the presence of 100 µg/mL Abs were compared to the control (no Ab) using ANOVA (V1 and V4) or Kruskal-Wallis (V2, V3, and V5). Pairwise t-test or

*Figure 3 continued on next page*

*Figure 3 continued*

Dunn significance is indicated by the symbols *, **, ***, and **** for adjusted p-values below 0.05, 0.01, 0.001, and 0.0001, respectively. The absence of a symbol corresponds to adjusted *p*-values >0.05. Data correspond to at least two experiments with three technical replicates each. Source Data: ***Figure 3—source data 1***.

The online version of this article includes the following source data and figure supplement(s) for figure 3:

**Source data 1.** Raw data for plots of ***Figure 3*** - Antibody inhibitory activity.

**Figure supplement 1.** Dose-dependent inhibition by mAbs P5B3 and P3D6 of ExoS-Bla injection from strains expressing five PcrV variants.

reduced the cytotoxicity induced by PAO1Δ3Tox by 28% and 73%, respectively (***Figure 4A***). Monoclonal Ab P3D6 exhibited a dose-response inhibition with an estimated $IC_{50}$ of 11.8 µg/mL (***Figure 4B***), while P5B3 did not exhibit a significant dose-response effect at concentrations below 100 µg/mL (***Figure 4C***). Overall, these results indicate that the binding of both mAbs to PcrV reduces the formation of the translocation pore in target cell membranes, with P3D6 exhibiting more potent activity.

## Crystal structure of PcrV* bound to Fab P3D6

In order to identify the PcrV epitopes recognized by the two mAbs, we generated a plasmid encoding a form of PcrV (PcrV*) amenable to crystallization (***Tabor et al., 2018***) as well as Fab fragments from both P3D6 and P5B3 mAbs. PcrV* was expressed in *E. coli,* while both Fabs were expressed in HEK293F cells. Individual proteins were purified by affinity and size-exclusion chromatographies. PcrV* was incubated with either Fab fragment, and samples were co-purified using size exclusion chromatography. Despite the fact that both PcrV*-Fab P3D6 and PcrV*-Fab P5B3 complexes co-eluted in gel filtration, only the PcrV*-Fab P3D6 complex subsequently generated diffracting crystals. Data were collected at the ESRF synchrotron in Grenoble, and the structure was solved by molecular replacement using Phaser (***McCoy et al., 2007***). Iterative manual model building and model improvement led to the structure whose statistics for data collection and refinement are presented in ***Supplementary file 4***.

PcrV* is composed of six helices interwoven by loop regions. α-helices 1, 4, and 6 are the major secondary structure elements in PcrV*, while helices 2, 3, and 5 are 1- or 2-turn helices. Most of the contacts formed between PcrV* and Fab P3D6 involve Helix 6 and the loop preceding it (***Figure 5A and B***) and implicate a binding platform made by both LC and HC from Fab P3D (***Figure 5—figure supplement 1***). From the PcrV side, the interaction region is highly polar, being formed by the side chains of Lys208, Gln217, Glu220, Lys222, Ser225, Asp226, Tyr228, Glu231, Asn234, Thr243, Asp246, and Arg247. The substitution of Ser225 in PcrV variant V1 by Lys or Arg in variants V2 to V5 is consistent with P3D6 being inefficient on strains harboring these four variants, since a residue with a bulky side chain in this position would invariably clash with the loop formed by residues 52–56 of the Fab.

In order to understand the protective role of mAb P3D6 in the context of the PcrV pentameric oligomer located at the tip of the PscF needle, we generated a model using the cryo-EM structure of the SipD pentamer (***Guo and Galán, 2021***) and aligned our co-crystal structure onto this model (***Figure 5C and D***). This analysis revealed that Fab P3D6 can successfully bind to one PcrV monomer (dark red and orange in ***Figure 5***, respectively), but would be unable to bind to a pre-formed PcrV pentamer due to the generation of clashes with neighboring subunits of the oligomeric form (***Figure 5***; the structure of the Fab can be seen overlaid with that of the pentamer subunits).

## Discussion

We generated a panel of anti-PscF and -PcrV human mAbs through specific memory B cell sorting from selected individuals. Although adsorption experiments with recombinant PscF suggested the presence of anti-PscF Abs with T3SS inhibitory activity in the donor from whom they were isolated, none of the isolated mAbs exhibited this activity. Competition mapping showed that the anti-PscF mAbs targeted three distinct regions of PscF, none of which were seemingly involved in inhibitory activity. Further epitope mapping would be necessary to gain deeper insight; however, in the absence of a PscF structure, this remains challenging. Isolating a greater number of mAbs from a selected donor with strong anti-PscF inhibitory activity would certainly increase the likelihood of identifying one with T3SS-inhibiting properties.

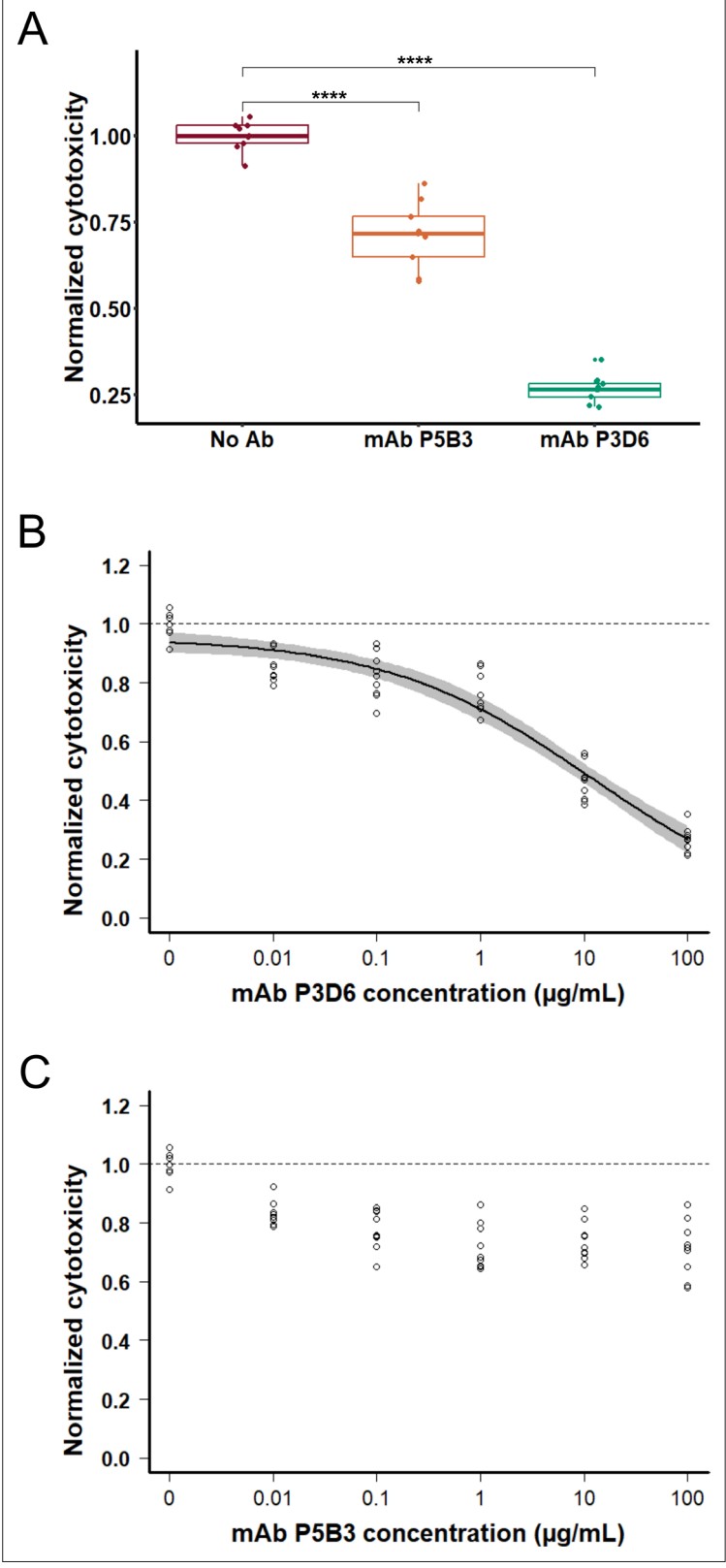

**Figure 4.** Monoclonal antibody (mAb) P3D6 efficiently inhibits the PopB/PopD translocation pore. J774 macrophages were infected with *P. aeruginosa* strain (PAO1, V1) deprived of all three T3SS toxins. The cell death (cytotoxicity) resulting from insertion of the translocon pore was measured by propidium iodide incorporation and normalized to the wild-type strain without addition of mAbs. Data correspond to three experiments with

*Figure 4 continued on next page*

*Figure 4 continued*

three technical replicates each. (**A**) Normalized cytotoxicity values in the presence of mAb at 100 µg/mL. Specific mAbs were compared to the control (no Ab) using ANOVA. Pairwise t-test significance is indicated by the symbol ****, meaning p-values below 0.0001. (**B, C**) Dose-response analysis with mAb concentrations ranging from 0.01 to 100 µg/mL. The circles, the black line and the gray area represent the experimental values, the log-logistic modeled dose-response curve and the 95% confidence interval, respectively. The dashed lines represent the mean of normalized cytotoxicity in the absence of Ab. The Ab concentration is presented in logarithmic scale. No black curve nor gray area is displayed for P5B3 because no dose-response could be modeled. In contrast, P3D6 exhibits an $IC_{50}$ of 11.8 µg/mL. Source Data: *Figure 4—source data 1*.

The online version of this article includes the following source data for figure 4:

**Source data 1.** Raw data for plots of *Figure 4* - Antibody inhibitory activity.

Of the four anti-PcrV mAbs isolated, two exhibited T3SS inhibitory capacity. Their mechanism of action could potentially involve (i) prevention of effector secretion by acting as a cap for PcrV; (ii) prevention of effector translocation towards the host cell by disruption of the PcrV-PopB/PopD interaction; or (iii) prevention of oligomerization of PcrV itself (*Gébus et al., 2009*; *Sawa et al., 2019*). Here, we measured the ability of the inhibiting anti-PcrV mAbs we isolated to block PopB/PopD pore formation and toxin injection and carried out a mechanistic and structural analysis of their activity, in parallel with other mAbs targeting PcrV.

We set out to investigate and compare the mechanism of action of several anti-PcrV mAbs: P3D6 mAb and P5B3 (this work), 30-B8 (*Simonis et al., 2023*), as well as a previously reported humanized, bivalent PcrV-Psl mAb (*DiGiandomenico et al., 2014*). In order to do so, we produced mAb 30-B8 and subsequently purchased mAb MEDI3902 from MedChem. The P3D6, P5B3, 30-B8, and MEDI3902 mAbs were notably compared by employing two assays capable of detecting T3SS inhibition, each with a different readout: injection of the ExoS-Bla reporter into epithelial cells and cytotoxicity measurements in macrophages as a read-out for translocon assembly. In addition, we performed structural analyses on three of the mAb-PcrV complexes, from the viewpoints of recognition of both monomeric and pentameric forms of PcrV.

Both MEDI3902 and 30-B8 mAbs potently inhibited the injection of ExoS-Bla into target cells, with $IC_{50}$ values of 117 ng/mL and 21.3 ng/mL, respectively (*Figure 6A*). Monoclonal Abs isolated in this study also inhibited toxin injection, although significantly less potently, with $IC_{50}$ values of 3.65 µg/mL for P3D6, and around 100 µg/mL for P5B3. Monoclonal Ab MEDI3902 had previously been shown to bind to different PcrV variants (*Tabor et al., 2018*) and here, we confirmed that mAb 30-B8 was similarly efficient at inhibiting toxin injection by strains carrying five different PcrV variants (*Simonis et al., 2023*; *Figure 6—figure supplement 1*). Monoclonal Ab P5B3 was also able to inhibit all variants, but only at high concentrations, while P3D6 was only active against variant V1. Therefore, P5B3 appears to recognize a highly conserved epitope, whereas P3D6 seems to bind an overlapping epitope that includes the variable Ser225, as suggested by the ELISA mapping competition and structural data, but does so in a more effective manner.

In the macrophage cytotoxicity assay, translocon assembly was inhibited by mAbs 30-B8 ($IC_{50}$ of 45.2 ng/mL) and P3D6 ($IC_{50}$ of 11.8 µg/mL), while P5B3 and MEDI3902 did not exhibit significant dose-response inhibition. Together, these results suggest that the T3SS-inhibiting activity of these anti-PcrV Abs may occur through distinct mechanisms.

In order to understand the differences in potency and potentially in mechanisms of action at a structural level, we compared the Fab-PcrV interaction regions for the P3D6, 30-B8 and MEDI3902 Fabs. Monomeric PcrV is an elongated, dumbbell-shaped molecule, and the chimeric form used in this study displays the same characteristics (*Figure 5A*). Fab MEDI3902 binds to one of the extremities of monomeric PcrV, extending in the longitudinal axis of the molecule (*Figure 6B*, top). This mode of binding is distinct from that of 30-B8 and P3D6 Fabs, both of which recognize the C-terminal region of PcrV (*Figure 6B*, middle) (*Simonis et al., 2023*). Therefore, the binding region of 30-B8 and P3D6 does not, by itself, appear to explain the significant difference in potency between the two mAbs. Moreover, a difference in affinity does not account for the difference in potency either, as both mAbs bind to recombinant monomeric PcrV with comparable apparent affinities of around 30 ng/mL (our data and *Simonis et al., 2023*).

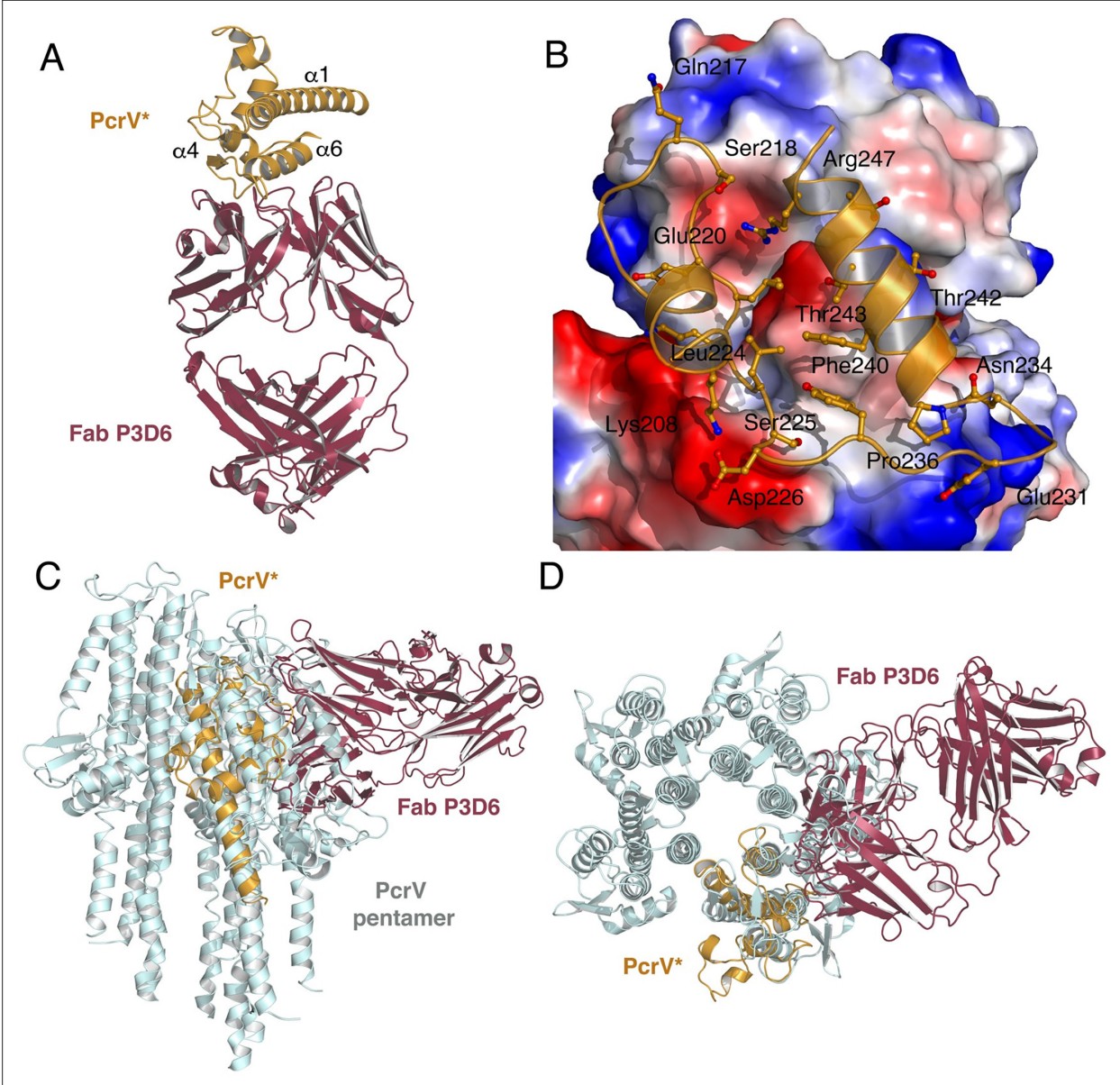

**Figure 5.** Structure of Fab P3D6 in complex with PcrV*. (**A**) Crystal structure of Fab P3D6 in complex with PcrV*. Fab P3D6 is shown in brown, while PcrV* is in orange. Contacts are made between PcrV* and an interaction platform formed by both HC and LC of P3D6. (**B**) Close-up of the interaction between PcrV* and P3D6, with the latter being shown as an electrostatic surface where acidic regions are shown in red, and basic in blue. Side (**C**) and top (**D**) views of the modeled PcrV pentamer, in light blue onto which the structure shown in (**A**) was overlaid.

The online version of this article includes the following figure supplement(s) for figure 5:

**Figure supplement 1.** Interactions between PcrV* and Fab P3D6.

In order to perform this comparative analysis in the context of a PcrV oligomer, we employed the model of the PcrV pentamer generated as described above, which was based on the cryo-EM structure of SipD from *S. typhimurium* (**Guo and Galán, 2021**). According to this analysis, the only Fab that is able to bind to all PcrV protomers in the pentamer without generating clashes with either PcrV or other Fabs is 30-B8 (**Figure 6**). In the case of MEDI3902, the Fab can bind the oligomer, but only with a 1:5 stoichiometry, possibly due to clashes between Fabs. Finally, P3D6 is unable to bind to a preformed pentamer and can only recognize a PcrV protomer in its monomeric form.

This structural analysis suggests different potential mechanisms of action. For P3D6, the inability to bind to the pentamer points towards a mechanism involving inhibition of oligomerization. Indeed,

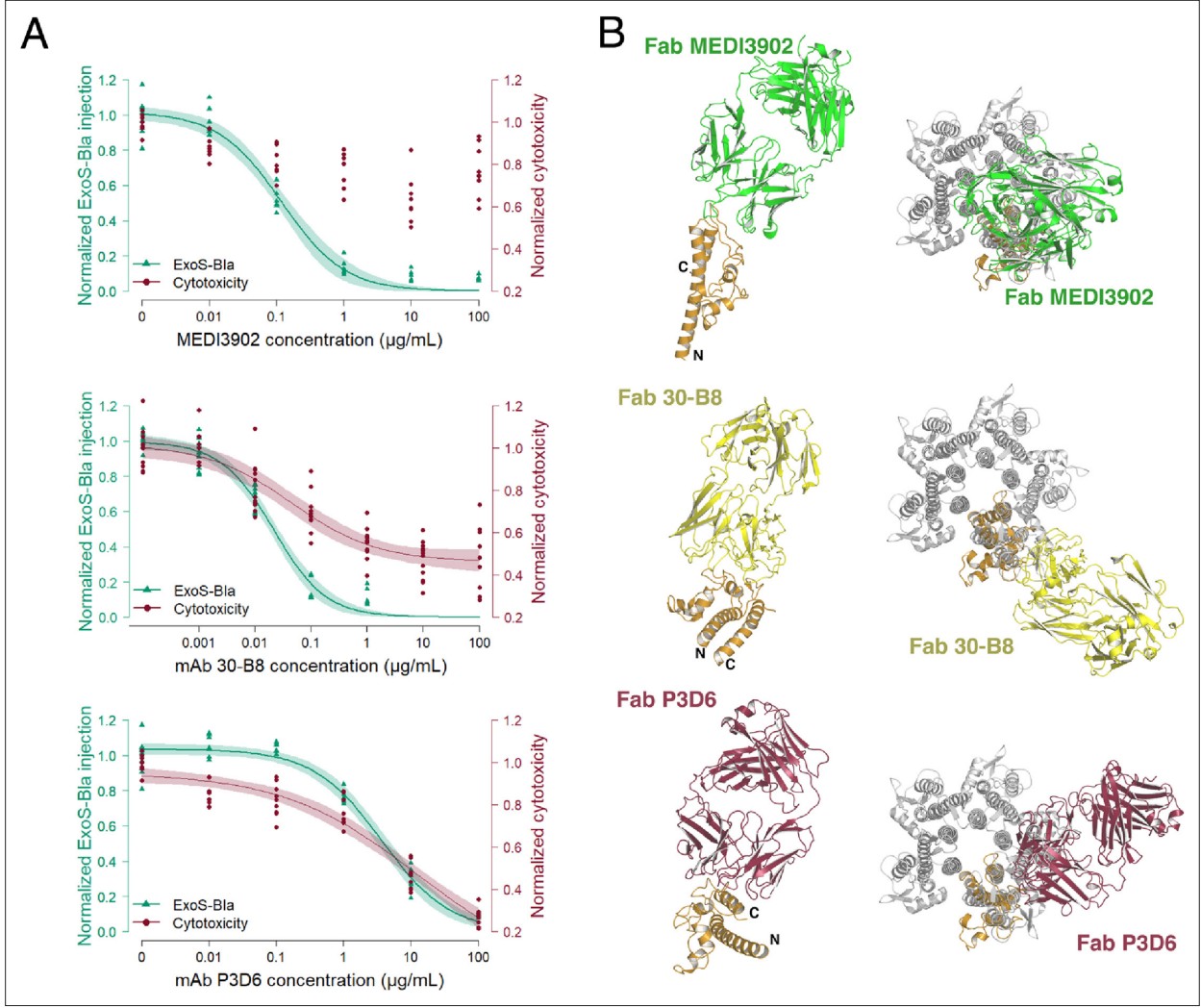

**Figure 6.** Functional and structural comparisons between anti-PcrV monoclonal antibodies (mAbs). (**A**) Dose-dependent inhibition of T3SS in two functional assays reflecting toxin injection (ExoS-Bla injection) and translocon assembly (J774 macrophage cytotoxicity) for three mAbs. Data correspond to three experiments with three technical replicates each. The circles, the dark lines, and the light-colored areas represent the experimental values, the log-logistic modeled dose-response curves and the 95% confidence intervals, respectively. The Ab concentration is presented in logarithmic scale. No red curve nor red area is displayed for MEDI3902 because no dose-response could be modeled. (**B**) Structures of PcrV-Fab complexes in the context of a PcrV monomer, as well as of a pentamer modeled based on the cryo-EM structure of SipD (*Guo and Galán, 2021*). The N- and C- termini of PcrV are indicated in all structures, which are referenced in the main text. Source Data: *Figure 6—source data 1*.

The online version of this article includes the following source data and figure supplement(s) for figure 6:

**Source data 1.** Raw data for plots of *Figure 6* - Antibody inhibitory activity.

**Figure supplement 1.** Dose-dependent inhibition by monoclonal antibodies (mAbs) 30-B8 of ExoS-Bla injection from strains expressing five PcrV variants.

the analysis shows that binding of P3D6 to a single PcrV monomer prevents the association of additional PcrV protomers through Fab-protomer direct clashes. A similar mechanism of action may be suggested for MEDI3902, as Fab-bound PcrV protomers cannot oligomerize, in this case due to clashes between the Fabs themselves (*Figure 6*). However, the fact that MEDI3902 does not appear to prevent translocon insertion suggests that such a mechanism is unlikely for this mAb as PcrV oligomerisation is required at the tip of the needle for translocon assembly. In contrast to P3D6, MEDI3902 can bind the oligomerized PcrV pentamer (with a stoichiometry of one), and its mechanism of action may thus be related to this ability. Our results suggest that when MEDI3902 is bound to the PcrV oligomer, the formation of the pore is not efficiently blocked while toxin injection is strongly inhibited (*Figure 6*). Therefore, the presence of one MEDI3902 Ab molecule at the tip of the needle does not

appear to efficiently prevent either the secretion of the translocator proteins PopB/PopD through the needle or the interactions between PcrV and PopB/PopD, which have been described as required for pore formation (*Kundracik et al., 2022*). However, the MEDI3902 Ab molecule seems to interfere with further interactions between PcrV and the PopB/PopB complex, or with the sensing by PcrV of the host cell, both of which are needed for toxin injection (*Lee et al., 2010*).

Lastly, 30-B8, which can bind the formed pentamer with a stoichiometry of five, appears to be the most effective at blocking both pore formation and toxin injection. The fact that the PcrV-bound 30-B8 Ab probably projects towards the cell membrane, associated with its ability to bind to the PcrV pentamer at full occupancy, may result in remarkable efficacy in blocking the interactions between PcrV and PopB/PopD.

In conclusion, here, we show that patients with chronic infection with *P. aeruginosa* can elicit anti-PscF and anti-PcrV mAbs that recognize different regions within these proteins. Anti-PcrV Abs can act as T3SS inhibitors through different mechanisms, with some exhibiting significantly greater efficacy than others. The strategy employed here, involving the analysis of structural and functional data on anti-T3SS mAbs should open new avenues towards deciphering the mechanism of T3SS toxin translocation and enable the isolation of more effective mAbs targeting a broad range of clinical strains.

# Materials and methods

**Key resources table**

| Reagent type (species) or resource | Designation | Source or reference | Identifiers | Additional information |
|---|---|---|---|---|
| Strain, strain background (*Escherichia coli*) | BL21(DE3) | Thermo Fisher | #EC0114 | Production of recombinant proteins |
| Strain, strain background (*Pseudomonas aeruginosa*) | CHAΔexoS::exoS-bla | PMID:22299042 | RRID:NCBITaxon_136841 | T3SS functionality assays |
| Strain, strain background (*P. aeruginosa*) | CHAΔpcrV | PMID:15271936 | RRID:NCBITaxon_136841 | T3SS functionality assays |
| Strain, strain background (*P. aeruginosa*) | PAO1Δ3Tox | PMID:18039770 | RRID:NCBITaxon_136841 | T3SS functionality assays |
| Cell line (mouse) | J774A.1, macrophages | ATCC | TIB-67; RRID:CVCL_0358 | mycoplasma-free, authenticated by Short Tandem Repeat (STR) profiling by Eurofins Genomics |
| Cell line (human) | A-549, lung epithelial cells | ATCC | CCL-185; RRID:CVCL_0023 | mycoplasma-free, authenticated by Short Tandem Repeat (STR) profiling by Eurofins Genomics |
| Cell line (human) | HEK293-F | Thermo Fisher Scientific | #R79007; RRID:CVCL_6642 | monoclonal antibody production, mycoplasma-free, authenticated by Short Tandem Repeat (STR) profiling by Eurofins Genomics |
| Biological sample (human) | Sera and PBMC | This work | | Approved by French ethics committee (ID-RCB 2020A00311-38), screening of patients' sera (dilution 1:50) and memory B cells, sera available from IBS, Grenoble |
| Antibody | Patients' purified IgGs (human polyclonal) | This work | | Used: 40–160 µg/mL, available from IBS, Grenoble |
| Antibody | VRCO1 (human monoclonal) | PMID:20616233 | | Produced during this work based on the published sequence, used: 100 µg/mL |
| Antibody | P5B3 (human monoclonal) | This work | | Used: 0.001–100 µg/mL, sequence in ***Supplementary file 6*** |
| Antibody | P3D6 (human monoclonal) | This work | | Used: 0.001–100 µg/mL, sequence in ***Supplementary file 6*** |

*Continued on next page*

*Continued*

| Reagent type (species) or resource | Designation | Source or reference | Identifiers | Additional information |
|---|---|---|---|---|
| Antibody | 30-B8 (human monoclonal) | PMID:37918395 | | Produced during this work based on the published sequence, used: 0.001–100 μg/mL |
| Antibody | MEDI3902, (human monoclonal) | Proteogenix | #PX-TA1591 | Used: 0.001–100 μg/mL |
| Antibody | Anti-PcrV, (rabbit polyclonal) | PMID:15271936 | | Controls in ELISA, used: 0.001–100 μg/mL |
| Antibody | Anti-PscF, (rabbit polyclonal) | PMID:15271936 | | Controls in ELISA, used: 0.001–100 μg/mL |
| Antibody | Anti-rabbit AP-coupled (goat polyclonal) | Thermo Fisher Scientific | #65–6122; RRID:AB_2533968 | ELISA (1:10000 dilution) |
| Antibody | Anti-human AP-coupled (goat polyclonal) | Jackson ImmunoResearch Labs | #109-056-098; RRID:AB_2337618 | ELISA (1:10000 dilution) |
| Antibody | Anti-human CD3 VioBlue (human monoclonal) | Miltenyi | #130–114- 519; RRID:AB_2726687 | Sorting of specific memory B cells |
| Antibody | Anti-human CD20 PE- Vio 770 (human monoclonal) | Miltenyi | #130–111- 340; RRID:AB_2656074 | Sorting of specific memory B cells |
| Antibody | Anti-human CD19 PE- Vio 770 (human monoclonal) | Miltenyi | #130–113- 647; RRID:AB_2726200 | Sorting of specific memory B cells |
| Antibody | Anti-human IgM PE (mouse monoclonal) | Miltenyi | #130–093- 075; RRID:AB_1036088 | Sorting of specific memory B cells |
| Antibody | Anti-human IgA PE (mouse monoclonal) | Miltenyi | #130–113- 476; RRID:AB_2733861 | Sorting of specific memory B cells |
| Antibody | Anti-human IgD PE (human monoclonal) | Miltenyi | #130–110- 643; RRID:AB_2652262 | Sorting of specific memory B cells |
| Antibody | Anti-human CD27 APC (human monoclonal) | Miltenyi | #130–113- 636; RRID:AB_2751162 | Sorting of specific memory B cells |
| Recombinant DNA reagent | pIApG-pcrV-V1 (PAO1) (plasmid) | This work | | Replicative plasmid for PcrV expression, *Leu6Ala9Ser21Ser225, available from IBS, Grenoble |
| Recombinant DNA reagent | pIApG-pcrV-V2 (CHA) (plasmid) | This work | | Leu6Ala9Ser21Arg225, available from IBS, Grenoble |
| Recombinant DNA reagent | pIApG-pcrV-V3 (PA14) (plasmid) | This work | | Phe6Ala9Pro21Lys225, available from IBS, Grenoble |
| Recombinant DNA reagent | pIApG-pcrV-V4 (PA103) (plasmid) | This work | | Phe6Gly9Pro21Arg225, available from IBS, Grenoble |
| Recombinant DNA reagent | pIApG-pcrV-V5 (plasmid) | This work | | Phe6Gly9Pro21Lys225, available from IBS, Grenoble |
| Recombinant DNA reagent | pET15b-His-PcrV (plasmid) | PMID:14565848 | | PcrV production, available from IBS, Grenoble |
| Recombinant DNA reagent | pET22b-PscF-His (plasmid) | PMID:16115870 | | PscF production, available from IBS, Grenoble |
| Recombinant DNA reagent | pESPRIT-His-PcrV-avitag (plasmid) | this work | | Production of PcrV-avitag for B cell sorting, available from IBS, Grenoble |
| Recombinant DNA reagent | pESPRIT-His-PscF-avitag (plasmid) | This work | | Production of PscF-avitag for B cell sorting, available from IBS, Grenoble |
| Recombinant DNA reagent | pET15b-PcrV* (plasmid) | This work | | Production of PcrV* containing amino acids (1-17)(136-249), available from IBS, Grenoble |

*Continued on next page*

*Continued*

| Reagent type (species) or resource | Designation | Source or reference | Identifiers | Additional information |
|---|---|---|---|---|
| Recombinant DNA reagent | Variable domains of heavy and light chains cloned into gamma1 HC, kappa LC, and lambda LC expression vectors | This work, PMID:17996249 | | Sequences provided in **Supplementary file 6**, available from IBS, Grenoble |
| Chemical compound, drug | Propidium Iodide | Sigma | #P4864 | |
| Chemical compound, drug | Aqua LIVE/DEAD stain | Thermo Fisher Scientific | #L34957 | Sorting of specific memory B cells |
| Chemical compound, drug | Streptavidin BUV737 | BD | #612775; RRID:AB_2869560 | Sorting of specific memory B cells |
| Chemical compound, drug | Streptavidin Vio-515 | Miltenyi | #103-107-459 | Sorting of specific memory B cells |
| Chemical compound, drug | Streptavidin BUV496 | BD | #612961; RRID:AB_2869599 | Sorting of specific memory B cells |
| chemical compound, drug | Streptavidin BV605 | Biolegend | #405229; RRID:AB_2869476 | Sorting of specific memory B cells |
| Chemical compound, drug | protein inhibitor cocktail | Roche | #4693132001 | Protein purification |
| Chemical compound, drug | ni-IDA resin | Macherey-Nagel | #745210–120 | Protein purification |
| Chemical compound, drug | 293 fectin | Fisher Scientific | #10553283 | Transfection reagent for mAb expression |
| Chemical compound, drug | SAX biosensors | Sartorius | #18–5,117 | BLI experiments |
| Chemical compound, drug | SA biosensors | Sartorius | #18–5019 | BLI experiments |
| Chemical compound, drug | CCF2 | Invitrogen | K1039 | Screening of functional antibodies |
| Chemical compound, drug | FreeStyle 293 F | Fisher Scientific | #10319322 | Medium for HEK293F, monoclonal antibody production |
| Commercial assay or kit | Quickchange II | Agilent | #200524 | Site-directed mutagenesis |

## Clinical sample collection

The study was approved by the French ethics committee (ID-RCB 2020A00311-38) and was carried out according to the Declaration of Helsinki, Good Clinical Practice (GCP) guidelines, and current French regulations. Written consent for participation was not required for this study. The first phase was a non-interventional study involving data and samples from human participants conducted according to Reference Methodology No. 004 issued by French authorities (Commission Nationale de l'Informatique et des Libertés). Screening and functional assays were performed on human sera previously collected at Grenoble Alpes University Hospital (France), from CF patients chronically infected with *P. aeruginosa*. Participants were all informed and did not object to this phase of the study. Inclusion criteria for the second phase of the study were: patients with positive screening during phase one, ≥18 years old, ≥32 Kg, with a programmed blood sampling at Grenoble Alpes University Hospital and not being opposed to the second phase of the project. Whole blood was then collected using BD Vacutainer EDTA tubes (Becton Dickinson) and PBMCs were purified by density gradient centrifugation using Lymphoprep (Eurobio Scientific) following manufacturing guidelines. Cells were then stored in liquid nitrogen until further use.

## Bacterial strains, genetic manipulations, and growth conditions

*P. aeruginosa* strains used in this study are listed in **Supplementary file 5**. Strains were cultured in LB media at 37 °C. For infection experiments, bacteria were grown until the measured optical density at

600 nm ($OD_{600nm}$) of 1. Genes encoding the most common variants of PcrV were cloned into the *Pseudomonas* replicative vector derived from pUCP21 (*West et al., 1994*) containing the P*pcrG* promoter that drives *pcrGVHpopDB* operon expression. The plasmids, kindly provided by Simona Barzu (Sanofi Pasteur, Lyon), were transformed into the *P. aeruginosa* strain CHA lacking *pcrV* (*Goure et al., 2004*).

### Cell lines
Three cell lines have been used in this work: HEK293F (Thermo Fisher Scientific) for monoclonal antibody production, and A549 (ATCC, CCL-185) and J774A.1 (ATCC, TIB-67) for bacterial cytotoxicity assays. They were tested mycoplasm-free and authenticated by Short Tandem Repeat (STR) profiling by Eurofins Genomics.

### Expression and purification of full-length PcrV and PcrV*
Expression of full-length PcrV from strain PAO1, cloned into a pET15b vector, was performed in *E. coli* BL21(DE3) as previously described, with small modifications (*Nanao et al., 2003*). Expression was induced with 1 mM of Isopropyl β-D-1-thiogalactopyranoside (IPTG) at $OD_{600}$=0.8 AU and cells were then grown overnight at 20 °C with shaking at 250 rpm. Cells were harvested by centrifugation and lysed by passing through a French Press three times at 25 Kpsi in lysis buffer (50 mM Tris pH 8, 200 mM NaCl, 20 mM Imidazole) supplemented with a protein inhibitor cocktail tablet (Roche). The supernatant was cleared by centrifugation at 18,000 rpm and subsequently loaded onto Ni-IDA resin (Macherey-Nagel). The resin was washed with lysis buffer, and the sample was eluted with lysis buffer supplemented with 100 mM imidazole. Fractions containing the sample were pooled and applied to a size exclusion chromatography column (Superdex 200 HiLoad 16/600) pre-equilibrated in SEC buffer (20 mM Tris pH 8, 150 mM NaCl, 1 mM EDTA). A chimeric form of PcrV (PcrV*) consisting of amino acids 1–17 fused to 136–249 whose design was inspired by the construct described in *Tabor et al., 2018*, was employed for crystallization purposes. The purification protocol was the same as above, the only difference being that 250 mM imidazole was employed to elute the sample from the Ni resin.

### Expression and purification of PscF
Expression of PscF from strain PAO1 was performed in *E. coli* BL21(DE3) grown in Terrific Broth. Expression was induced with 1 mM IPTG at $OD_{600}$=0.6 AU and cells were then grown for an additional 3 hr at 37 °C with shaking at 250 rpm. Cells were harvested by centrifugation and lysed by passing through a French Press three times at 25 kpsi in lysis buffer (50 mM Tris pH 8, 200 mM NaCl, 20 mM Imidazole, 2% glycerol) supplemented with a protein inhibitor cocktail tablet (Roche). The supernatant was cleared by centrifugation at 18,000 rpm and applied to a Ni-IDA resin (Macherey-Nagel). The resin was washed with lysis buffer, and the protein was eluted in the same buffer supplemented with 250 mM imidazole. Fractions were then buffer exchanged in an Amicon Ultra 10 kDa cutoff concentrator against a buffer exempt of imidazole (50 mM Tris pH 8, 200 mM NaCl, 2% glycerol).

### ELISA assays
For direct ELISA, 96-well ELISA plates (Fisher # 11530627, Nunc Maxisorp) were coated overnight at 4 °C with the respective antigen diluted to 1 µg/ml in PBS. Plates were then washed with PBS-Tween 0.01%, and blocked for 1 hr at room temperature (RT) with 3% BSA in PBS. Next, sera or mAbs serially diluted in PBS-BSA 1% were added and incubated for 1 hr at RT. Antibody binding was detected using alkaline phosphatase-coupled goat anti-human IgG (Jackson Immuno #109 056 098) and a para-nitrophenylphosphate substrate (Interchim #UP 664791). The enzymatic reaction was read at 405 nm using a TECAN Spark 10 M plate reader. Polyclonal Abs raised in rabbits against PscF and PcrV (*Goure et al., 2004*) were used as positive controls.

For competition ELISAs, serial dilutions of competitor mAbs were transferred into antigen-coated wells. Following a 30 min incubation, biotinylated mAbs were added to the wells at their $EC_{70}$ concentration (effective concentration for 70% binding). Binding of biotinylated mAbs was detected using alkaline phosphatase-conjugated streptavidin.

### Sorting of specific memory B cells
Briefly, PBMCs were stained for 30 min at 4 °C in the dark, using Facs-Buffer (PBS-1X0.5% BSA, 2 mM EDTA) with Live Dead staining (Thermo L34957), Antihuman CD3-Vio-Blue (Miltenyi 130-114-519),

Anti-human CD20 Pe-Vio707 (Miltenyi 130-111-345), Anti-human CD19 Pe-Vio707 (Miltenyi 130-113-649), Anti-human IgM PE (Miltenyi 130-093-075), Anti-human IgA PE (Miltenyi 130-113-476), Anti-human IgD PE (Miltenyi 130-110-643), Anti-human CD27 APC (Miltenyi 130-108-336), in the presence of recombinant biotinylated His-PcrV-Avitag coupled with streptavidin BUV737 (BD 612775) or streptavidin Vio-515 (Miltenyi 103-107-459), and recombinant biotinylated His-PscF-Avitag coupled with streptavidin BUV496 (BD 612961) or streptavidin BV605 (Biolegend 405229). After washing, the cells were resuspended in FACS-Buffer and PscF or PcrV positive B cells were sorted and clonally seeded in 96 plates containing lysis buffer using BD FACSAria Fusion cytometer (BD Biosciences).

## Isolation and production of mAbs

Sequences coding for variable regions of both heavy and light ($\kappa$ and $\lambda$) chains were isolated by reverse transcription on total mRNA followed by a multiplex nested PCR using a set of primers (*Tiller et al., 2008*) covering the diversity of V-region diversity. The V-regions family was attributed after sequencing of amplicons and alignment in the IMGT database (https://imgt.org/). An additional round of PCR using primers specific to the identified family (*Tiller et al., 2008*) was performed followed by the cloning of V-regions genes into corresponding vectors containing IgG1H, IgG$\kappa$, and IgG$\lambda$ constant regions. Sequences are provided in *Supplementary file 6*. Regarding 30-B8, the sequences coding for the variable regions of heavy and light chains were synthesized by Eurofins according to the sequence published by Simonis and coworkers (*Simonis et al., 2023*).

Monoclonal Abs were produced by transient transfection in HEK293F cells (Thermo Fisher Scientific) and purified by affinity chromatography using a Protein A Sepharose column (Sigma #GE17-1279-03). Elution was performed with 4.5 ml of glycine 0.1 M (pH 2.5), followed by neutralization with 500 µl of 1 M Tris (pH 9). Purified mAbs were then subjected to buffer exchange and concentration using Amicon Ultra centrifugal filters (Merck #36100101).

## P3D6 and P5B3 Fab production

Sequences coding for Fab fragments were obtained by inserting stop codons on genes corresponding to heavy chains of the mAbs by PCR using site-directed mutagenesis (Quickchange II, Agilent) according to the manufacturer's instructions. Mutated heavy and corresponding light chain genes were cloned into appropriate expression plasmids for eukaryotic cell expression and were co-transfected at a 2:1 ratio into FreeStyle 293 F cells (Thermo Fisher). Fabs were purified using KappaSelect affinity chromatography (Cytiva).

## Cellular tests for T3SS activity

### ExoS-Bla translocation

The T3SS-dependent toxin injection into epithelial A549 cells was measured using the reporter system based on Bla/CCF2 enzyme/substrate combination (*Charpentier and Oswald, 2004*) previously described for *P. aeruginosa* (*Verove et al., 2012*). *P. aeruginosa* strain CHA$\Delta exoS$ carrying ExoS-Bla fusion on the chromosome was used to infect A549 cells at the multiplicity of infection (MOI) of 5. The level of injected ExoS-β-lactamase was measured using CCF2 substrate, as described previously (*Verove et al., 2012*). Inhibition of ExoS-Bla translocation was evaluated in the presence of serial dilutions of Protein A-purified serum IgGs and mAbs. Serum IgG purification was performed as described above for mAb purification. All values were normalized using non-infected cells and cells infected in the absence of Abs as references.

### Pore formation/propidium iodide incorporation into macrophages

To assess the formation of a T3SS translocation pore, macrophages were infected with a *P. aeruginosa* strain PAO1$\Delta$3Tox devoid of three exotoxins (*Cisz et al., 2008*). Two days before the experiment, J774 cells were seeded in a 96-well plate (Greiner, 655090) at a density of 100,000 cells per well in Dulbecco's modified Eagle's medium (DMEM) supplemented with 10% FCS. The day before the experiment, the strain was grown overnight in LB medium. The next day, bacteria were sub-cultured in fresh LB media until an $OD_{600\ nm}$ of 1, and the macrophages were washed twice with PBS before addition of 65 µL of DMEM 10% FCS containing 2 µg/mL of propidium iodide. Antibodies diluted in DMEM with 10% FCS (25 µL) were then added, followed by 10 µL of bacteria diluted in DMEM 10% FCS to give a MOI of 5. Propidium iodide fluorescence was recorded in a Fluoroskan fluorimeter

every 10 min. The data from each fluorescence kinetics of the triplicates were processed in R Studio to calculate the Area Under the Curve, as described before (*Ngo et al., 2019*). This metric was then normalized using noninfected cells and cells infected in the absence of Abs as references.

## Data processing and analysis
Data from independent cell experiments were pooled and analyzed with R version 4.3.2 (*R Development Core Team, 2023*) by one-way ANOVA followed by paired t-test or Kruskal-Wallis followed by the Dunn test with Benjamini-Hochberg p-value adjustment. Dose-response fitting was performed using the drc package (*Ritz et al., 2015*) based on a three-parameter log-logistic model and $IC_{50}$ were compared using the function comParm().

## Bio-layer interferometry
BLI experiments were performed on an OctetRED96e from Satorius/FortéBio (former Pall/FortéBio) and were recorded with software provided by the manufacturer (Data Acquisition v11.1). All protein samples were diluted in analysis buffer (1 X PBS pH 7.4, 0.02% Tween-20). 10 mM glycine pH 2.0 was used as regeneration buffer. Commercial SA or SAX (streptavidin) biosensors (Pall/FortéBio) were used to capture biotinylated PcrV. Kinetic analyses were performed in black 96-well plates (Nunc F96 MicroWell, Thermo Fisher Scientific) at 25 °C with agitation at 1000 rpm. After incubation and equilibration of biosensors in analysis buffer, PcrV samples were applied at a concentration of 2.5 mg/mL by dipping biosensors until reaching a spectrum shift between 1.2 and 2 nm, followed by an additional equilibration step in analysis buffer. For association measurements, all analyte samples were diluted in analysis buffer at concentrations either between 3.12 and 200 nM for IgGs or between 50 and 3200 nM for Fab fragments. Association phases were monitored while dipping the functionalized biosensors in analyte solutions for 5 min after recording a baseline for 2 min, and the dissociation phases monitored in analysis buffer for 10 min. To assess and monitor unspecific binding of analytes, measurements were performed with biosensors treated with the same protocols but replacing ligand solutions with analysis buffer. All measurements were performed in duplicate using sample preparations. Kinetic data were processed with software provided by the manufacturer (Data analysis HT v11.1). Signals from zero-concentration samples were subtracted from the signals obtained for each functionalized biosensor and each analyte concentration. Resulting specific kinetics signals were then fitted using a global fit method and 1:2 bivalent analyte model for full Abs/IgG and 1:1 Langmuir model for Fab. Reported kinetics parameter values were obtained by averaging the values obtained with duplicated assays and reported errors as the standard deviation.

## Crystallization of the PcrV*-Fab P3D6 complex
PcrV* and Fab P3D6 were mixed in a 1:2 ratio for 1 hr at room temperature prior to being subjected to size exclusion chromatography using a Superdex 200 10/300 GL increase column in SEC buffer (20 mM Tris pH 7.4, 150 mM NaCl, 1 mM EDTA). Peaks harboring PcrV*:Fab complexes in SDS-PAGE were pooled, concentrated, and used for crystallization trials using the ISBG HTX crystallization platform in Grenoble. Initial crystallization conditions (25% PEG 1000, 1 mM $ZnCl_2$, 100 mM sodium acetate pH 5.5) were optimized manually, and diffracting crystals were obtained using microseeding. All crystals were grown using the hanging drop vapor diffusion method at 20 °C. Single crystals were mounted in cryo-loops and flash-cooled in liquid nitrogen. X-ray diffraction data were collected under a nitrogen stream at 100°K at the European Synchrotron Radiation Facility (ESRF, Grenoble, France).

## Structure determination and refinement
The best diffraction data were collected to 2.56 Å on beamline ID30A-1 (ESRF) (*Bowler et al., 2015*). The diffracting crystal was in space group $P2_1$ and displayed one 1:1 PcrV:Fab complex per asymmetric unit. Statistics on data collection and refinement are summarized in *Supplementary file 5*. X-ray diffraction images were indexed and scaled with XDS (*Kabsch, 2010*). ADXV (*Arvai, 2020*) and XDSGUI (*Brehm et al., 2023*) were used to perform data quality and resolution cutoff check-ups (*Karplus and Diederichs, 2015*). The maximum possible resolution was determined using the STARA-NISO server (*Tickle, 2007*). The reduced X-ray diffraction data was imported into the CCP4 program suite (*Agirre et al., 2023*). The PcrV*-Fab P3D6 structure was solved by molecular replacement using PHASER (*McCoy et al., 2007*) and an AlphaFold2 ColabFold-generated model (*Mirdita et al., 2022*).

The PcrV* and Fab model chains were placed sequentially. The structure was completed by cycles of manual model building with COOT (*Emsley and Cowtan, 2004*). Water molecules were added to the residual electron density map as implemented in COOT. Crystallographic macromolecular refinement was performed with REFMAC (*Murshudov et al., 2011*). Cycles of model building and refinement were performed until $R_{work}$ and $R_{free}$ converged. The TLS definition was determined and validated using the TLSMD (*Painter and Merritt, 2006*) and PARVATI (*Zucker et al., 2010*) servers. The stereochemical quality of the refined models was verified with MOLPROBITY (*Chen et al., 2010*), PROCHECK (*Laskowski et al., 1993*), and PDB-REDO (*Joosten et al., 2014*). Secondary structure assignment was performed by DSSP (*Kabsch and Sander, 1983*) and STRIDE (*Heinig and Frishman, 2004*). Figures displaying protein structures were generated with PYMOL (http://www.pymol.org).

## Acknowledgements

This work was supported by a grant from the Agence Nationale de la Recherche (ANR-22-CE18-0009) to PP, AD, and IA, as well as grant 183360 from the Région Auvergne Rhône-Alpes to PP and AD. This work used the platforms of the Grenoble Instruct-ERIC center (ISBG; UAR 3518 CNRS-CEA-UGA-EMBL) within the Grenoble Partnership for Structural Biology (PSB), supported by FRISBI (ANR-10-INBS-0005) and GRAL, financed within the University Grenoble Alpes graduate school (Ecoles Universitaires de Recherche) CBH-EUR-GS (ANR-17-EURE-0003). The IBS acknowledges integration into the Interdisciplinary Research Institute of Grenoble (CEA).

## Additional information

### Funding

| Funder | Grant reference number | Author |
| --- | --- | --- |
| Agence Nationale de la Recherche | 22-CE18-0009-02 | Ina Attree<br>Andrea Dessen<br>Pascal Poignard |
| Region Auvergne Rhones Alpes | 183360 | Andrea Dessen<br>Pascal Poignard |
| Agence Nationale de la Recherche | ANR-17-EURE-0003 | Ina Attree<br>Andrea Dessen<br>Pascal Poignard |
| Agence Nationale de la Recherche | ANR-10-INBS-0005-02 | Andrea Dessen |

The funders had no role in study design, data collection and interpretation, or the decision to submit the work for publication.

### Author contributions

Jean-Mathieu Desveaux, Conceptualization, Data curation, Formal analysis, Investigation, Methodology, Writing – original draft; Eric Faudry, Data curation, Formal analysis, Supervision, Visualization, Writing – review and editing; Carlos Contreras-Martel, Supervision, Investigation, Visualization; François Cretin, Supervision, Investigation, Methodology; Leonardo Sebastian Dergan-Dylon, Formal analysis, Supervision, Investigation, Visualization, Methodology; Axelle Amen, Fabien Chenavier, Investigation; Isabelle Bally, Delphine Fouquenet, Investigation, Methodology; Victor Tardivy-Casemajor, Investigation, Visualization; Yvan Caspar, Supervision, Methodology, Writing – review and editing; Ina Attree, Conceptualization, Formal analysis, Supervision, Funding acquisition, Writing – review and editing; Andrea Dessen, Formal analysis, Supervision, Funding acquisition, Writing – review and editing; Pascal Poignard, Conceptualization, Formal analysis, Supervision, Funding acquisition, Methodology, Project administration, Writing – review and editing

### Author ORCIDs

Eric Faudry ⓘ https://orcid.org/0000-0001-9958-6029
François Cretin ⓘ https://orcid.org/0000-0001-9939-0931

# eLife Research article

Microbiology and Infectious Disease

Leonardo Sebastian Dergan-Dylon https://orcid.org/0000-0002-5355-7107
Axelle Amen https://orcid.org/0000-0002-0449-4445
Isabelle Bally https://orcid.org/0000-0002-8315-6080
Victor Tardivy-Casemajor https://orcid.org/0009-0002-4490-8769
Fabien Chenavier https://orcid.org/0009-0004-6438-0716
Ina Attree https://orcid.org/0000-0002-2580-764X
Andrea Dessen https://orcid.org/0000-0001-6487-4020
Pascal Poignard https://orcid.org/0000-0002-0021-7192

### Ethics

Human subjects: Approved by the French ethics committee ID-RCB 2020A00311-38.

Reviewer #1 (Public review): https://doi.org/10.7554/eLife.105195.3.sa1
Reviewer #2 (Public review): https://doi.org/10.7554/eLife.105195.3.sa2
Author response https://doi.org/10.7554/eLife.105195.3.sa3

## Additional files

### Supplementary files

Supplementary file 1. Sequence conservation of V and J regions of selected mAbs compared to germline. Percentage (%) of identity was obtained by aligning variable region sequences on IMGT database (https://www.imgt.org/).

Supplementary file 2. Competition between (A) anti-PscF monoclonal antibodies (mAbs) and (B) anti-PscF mAbs. The indicated $IC_{50}$ values correspond to the concentration of competitor mAbs necessary to obtain half of the signal generated by the biotinylated mAbs without competitor. ND corresponds to a non-detectable competition. Source Data: *Source data 2*.

Supplementary file 3. Affinities of anti-PcrV monoclonal antibodies (mAbs) for PcrV. The reported values correspond to the average of the measurements obtained from two independent experiments (n=2). Standard Deviations were calculated by the BLI analysis software. Source Data: *Source data 2*.

Supplementary file 4. Data collection, phasing, and structure refinement statistics.

Supplementary file 5. Bacterial strains and plasmids.

Supplementary file 6. Antibody variable region sequences.

MDAR checklist

Source data 1. Raw data for tables of *Supplementary file 2* - Antibody competition.

Source data 2. Raw data for table of *Supplementary file 2* - Antibody affinities.

### Data availability

All data generated or analyzed during this study are included in the manuscript and supporting files, with the exception of the final refined model coordinates and structure factors corresponding to the PcrV*- Fab P3D6 complex. Those were deposited in the Protein Data Bank (PDB, https://www.rcsb.org), ID code: 9FM0. Antibody sequences are provided in *Supplementary file 6*.

The following dataset was generated:

| Author(s) | Year | Dataset title | Dataset URL | Database and Identifier |
|---|---|---|---|---|
| Desveaux JM, Contreras Martel C, Dessen A | 2025 | Human antibody (Fab) and *P. aeruginosa* (T3SS) protein PcrV-fragment complex | https://doi.org/10.2210/pdb9FM0/pdb | Worldwide Protein Data Bank, 10.2210/pdb9FM0/pdb |

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
