## [Editor Report · eLife Assessment]

This **useful** work identifies new monoclonal antibodies produced by cystic fibrosis patients against *Pseudomonas aeruginosa* type three secretion system. The evidence supporting authors' claim is **solid**. Nonetheless, the manuscript may benefit from a more in depth description of what the authors learned from their structure-based analyses of antibodies targeting PcrV.

---

## [Referee Report · Reviewer #1 (Public review)]

Summary:

Desveaux et al. describe human mAbs targeting protein from the *Pseudomonas aeruginosa* T3SS, discovered by employing single cell B cell sorting from cystic fibrosis patients. The mAbs were directed at the proteins PscF and PcrV. They particularly focused on two mAbs binding the T3SS with the potential of blocking activity. The supplemented biochemical analysis was crystal structures of P3D6 Fab complex. They also compared the blocking activity with mAbs that were described in previous studies, using an assay that evaluated the toxin injection. They conducted mechanistic structure analysis and found that these mAbs might act through different mechanisms by preventing PcrV oligomerization and disrupting PcrVs scaffolding function.

The antibiotic resistance crisis requires the development of new solutions to treat infections cause by MDR bacteria. The development of antibacterial mAbs holds great potential. In that context, this report is important as it paves the way for the development of additional mAbs targeting various pathogens that harbor the T3SS. In this report the authors present a comparative study of their discovered mAbs vs. a commercial mAb currently in clinical testing resulting in valuate data with applicative implications. The authors investigated the mechanism of action of the mAbs using advanced methods and assays for characterization of antibody and antigen interaction, underlining the effort to determine the discovered mAbs suitability for downstream application.

---

## [Referee Report · Reviewer #2 (Public review)]

Summary:

Desveaux et al. performed Elisa and translocation assays to identify among 34 cystic fibrosis patients which ones produced antibodies against *P. aeruginosa* type three secretion system (T3SS). Authors were especially interested in antibodies against PcrV and PcsF, two key components of the T3SS. The authors leveraged their binding assays and flow cytometry to isolate individual B cells from the two most promising sera, and then obtained monoclonal antibodies for the proteins of interest. Among the tested monoclonal antibodies, P3D6 and P5B3 emerged as the best candidates due to their inhibitory effect on the ExoS-Bla translocation marker (with 24% and 94% inhibition, respectively). The authors then showed that P5B3 binds to the five most common variants of PcrV, while P3D6 seems to recognize only one variant. Furthermore, the authors showed that P3D6 inhibits translocon formation, measured as cell death of J774 macrophages. To get insights into the P3D6-PcrV interaction, the authors defined the crystal structure of the P3D6-PcrV complex. Finally, the authors compared their new antibodies with two previous ones (i.e., MEDI3902 and 30-B8).

Strengths:

• Article is well written.

• Authors used complementary assays to evaluate protective effect of candidate monoclonal antibodies.

• Authors offered crystal structure with insights into the P3D6 antibody-T3SS interaction (e.g., interactions with monomer vs pentamers).

• Authors put their results in context by comparing their antibodies with respect to previous ones.

Weaknesses:

• Results shown in Fig. 6 should be initially described in the Results section and not in the Discussion section.

• The authors should describe, in the Discussion (and also in L146-147), in more detail the gained insights into how anti-PcrV antibodies work. This is especially important given previous reports of more potent antibodies (e.g., Simonis et al.) that significantly reduces the novelty of their work. Hence, authors could explicitly highlight how their study differentiate from previous work, and what unique insights were gained (in the current version is not completely obvious).

---

## [Author Response]

The following is the authors’ response to the original reviews.

**Reviewer #1 (Public review):**
Summary:Desveaux et al. describe human mAbs targeting protein from the *Pseudomonas aeruginosa* T3SS, discovered by employing single cell B cell sorting from cystic fibrosis patients. The mAbs were directed at the proteins PscF and PcrV. They particularly focused on two mAbs binding the T3SS with the potential of blocking activity. The supplemented biochemical analysis was crystal structures of P3D6 Fab complex. They also compared the blocking activity with mAbs that were described in previous studies, using an assay that evaluated the toxin injection. They conducted mechanistic structure analysis and found that these mAbs might act through different mechanisms by preventing PcrV oligomerization and disrupting PcrVs scaffolding function.Strengths:The antibiotic resistance crisis requires the development of new solutions to treat infections caused by MDR bacteria. The development of antibacterial mAbs holds great potential. In that context, this report is important as it paves the way for the development of additional mAbs targeting various pathogens that harbor the T3SS. In this report, the authors present a comparative study of their discovered mAbs vs. a commercial mAb currently in clinical testing resulting in valuable data with applicative implications. The authors investigated the mechanism of action of the mAbs using advanced methods and assays for the characterization of antibody and antigen interaction, underlining the effort to determine the discovered mAbs suitability for downstream application.Weaknesses:Although the information presented in this manuscript is important, previous reports regarding other T3SS structures complexed with antibodies, reduce the novelty of this report. Nevertheless, we provide several comments that may help to improve the report. The structural analysis of the presented mAbs is incomplete and unfortunately, the authors did not address any developability assessment. With such vital information missing, it is unclear if the proposed antibodies are suited for diagnostic or therapeutic usage. This vastly reduces the importance of the possibly great potential of the authors' findings. Moreover, the structural information does not include the interacting regions on the mAb which may impede the optimization of the mAb if it is required to improve its affinity.

As described in the manuscript (Fig. 6), our mAbs are markedly less effective in every in vitro T3SS inhibition assay than the mAbs recently described by Simonis et al. They are therefore very unlikely to outperform these mAbs in in vivo animal models of *P. aeruginosa* infection. Considering the high cost of animal experiments and ethical concerns-and in accordance with the Reduction principal of the 3Rs guidelines-we chose not to pursue in vivo experiments. Instead, we focused on leveraging the new isolated mAbs to investigate the mechanisms of action and structural features of anti-PcrV mAbs.

Following the reviewer's suggestion, we have now added mAb interaction features into the structural data presented in the manuscript. However, based on the efficiency data, the structural analysis and the mechanistic insights presented, we do not consider further therapeutic use and optimization of our mAbs to be warranted.

**Reviewer #2 (Public review):**
Summary:Desveaux et al. performed Elisa and translocation assays to identify among 34 cystic fibrosis patients which ones produced antibodies against *P. aeruginosa* type three secretion system (T3SS). The authors were especially interested in antibodies against PcrV and PcsF, two key components of the T3SS. The authors leveraged their binding assays and flow cytometry to isolate individual B cells from the two most promising sera, and then obtained monoclonal antibodies for the proteins of interest. Among the tested monoclonal antibodies, P3D6 and P5B3 emerged as the best candidates due to their inhibitory effect on the ExoS-Bla translocation marker (with 24% and 94% inhibition, respectively). The authors then showed that P5B3 binds to the five most common variants of PcrV, while P3D6 seems to recognize only one variant. Furthermore, the authors showed that P3D6 inhibits translocon formation, measured as cell death of J774 macrophages. To get insights into the P3D6PcrV interaction, the authors defined the crystal structure of the P3D6-PcrV complex. Finally, the authors compared their new antibodies with two previous ones (i.e., MEDI3902 and 30-B8).Strengths:(1) The article is well written.(2) The authors used complementary assays to evaluate the protective effect of candidate monoclonal antibodies.(3) The authors offered crystal structure with insights into the P3D6 antibody-T3SS interaction (e.g., interactions with monomer vs pentamers).(4) The authors put their results in context by comparing their antibodies with respect to previous ones.Weaknesses:The authors used a similar workflow to the one previously reported in Simonis et al. 2023 (antibodies from cystic fibrosis patients that included B cell isolation, antibody-PcrV interaction modeling, etc.) but the authors do not clearly explain how their work and findings differentiate from previous work.

We employed a similar mAb isolation pipeline to that used by Simonis et al., beginning with the screening of a cohort of cystic fibrosis patients chronically infected with *P. aeruginosa*. As in Simonis et al., we isolated specific B cells using a recombinant PcrV bait, followed by single-cell PCR amplification of immunoglobulin genes. The main differences in methodology between the two studies are as follows: (i) the use of individuals from different cohorts, and therefore having different Ab repertoires; (ii) the nature of the screening assays, although in both cases the screening was focused on the inhibition of T3SS function; (iii) the PcrV labeling strategy, with Simonis et al. employing direct labeling, whereas we used a biotinylated tag combined with streptavidin;

The number of specific mAbs obtained and produced was higher in Simonis et al. (47 versus 9 in our study). They sorted B cells from three individuals compared to two in our work and possibly started with a larger amount of PBMCs per donor, which may account for the higher number of specific B cells and mAbs isolated. Considering that the strategies were overall very similar, the greater number of mAbs isolated in Simonis et al. likely explains, to a large extent, why they identified mAbs targeting different epitopes compared to ours, including highly potent mAbs that we did not recover.

Our modeling study, unlike that of Simonis et al., which relied on an AlphaFold prediction of the multimeric structure of *P. aeruginosa* PcrV, was based on the experimentally determined structure of the homologous Salmonella SipD pentamer, as described in the manuscript. Furthermore, we compared our mAb P3D6 not only with 30-B8 from Simonis et al., but also with MEDI3902. Finally, in contrast to the approach of Simonis et al., we used functional assays to investigate the differences in mechanisms of action among these mAbs, which target three distinct epitopes.

(2) Although new antibodies against *P. aeruginosa* T3SS expand the potential space of antibodybased therapies, it is unclear if P3D6 or P5B3 are better than previous antibodies. In fact, in the discussion section authors suggested that the 30-B8 antibody seems to be the most effective of the tested antibodies.

As explained above and shown in the Results section (Figure 6), the 30-B8 mAb is markedly more effective at inhibiting T3SS activity in both in vitro assays used.

(3) The authors should explain better which of the two antibodies they have discovered would be better suited for follow-up studies. It is confusing that the authors focused the last sections of the manuscript on P3D6 despite P3D6 having a much lower ExoS-Bla inhibition effect than P5B3 and the limitation in the PcrV variant that P3D6 seems to recognize. A better description of this comparison and the criteria to select among candidate antibodies would help readers identify the main messages of the paper.

The P3D6 mAb shows stronger inhibitory activity than P5B3 in the two assays used, as shown in Supplementary Figure 1. An error in the table in Figure 2B was corrected and this table now reflects the results presented in Supplementary Figure 1.

The final sections of the manuscript focus on P3D6, which is more potent than P5B3, and for which we successfully determined a co-crystal structure with PcrV*. All parallel attempts to obtain a structure of P5B3 in complex with PcrV* failed. The P3D6-PcrV* structure was used to analyze epitope recognition and mechanisms of action in comparison to previously described mAbs. As previously mentioned, we do not consider further studies aimed at therapeutic development and optimization of our mAbs to be justified given the current data. Therefore, we believe that the main message of the paper is adequately captured in the title.

(4) This work could strongly benefit from two additional experiments:(a) In vivo experiments: experiments in animal models could offer a more comprehensive picture of the potential of the identified monoclonal antibodies. Additionally, this could help to answer a naïve question: why do the patients that have the antibodies still have chronic *P. aeruginosa* infections?

As explained above, the mAbs we isolated are significantly less potent than those described by Simonis et al., and are therefore unlikely to outperform the best anti-PcrV candidates in vivo. In light of the data, and considering ethical concerns related to animal use in research and budgetary constraints, we decided not to proceed with in vivo experiments.

There are a number of reasons that may explain why patients with anti-PcrV Abs blocking the T3SS can still be chronically infected with Pa. First these Abs may be at limiting concentration, particularly in sites where Pa replicates, and thus unable to clear infection. in addition, it has been described that the T3SS is downregulated in chronic infection in cystic fibrosis patients. This suggests that a therapeutic intervention with T3SS inhibiting Abs may be more efficient if done early in cystic fibrosis patients to prevent colonization when Pa possesses an active T3SS. Finally, T3SS is not the only virulence mechanism employed by *P. aeruginosa* during infection. Indeed, multiple protein adhesins and polysaccharides are important factors facilitating the formation of bacterial biofilms that are crucial for establishing chronic persistent infection. In this regard, a combination of Abs targeting different factors on the *P. aeruginosa* surface may be needed to treat chronic infections.

(b) Multi-antibody T3SS assays (i.e., a combination of two or more monoclonal antibodies evaluated with the same assays used for characterization of single ones). This could explore the synergistic effects of combinatorial therapies that could address some of the limitations of individual antibodies.

Given the high potency of the Simonis mAbs and the mechanisms of action highlighted by our analysis, it is unlikely that our mAbs would synergize with those described by Simonis. Additionally, since our two mAbs cross-compete for binding, synergy between them is also improbable.

**Reviewer #1 (Recommendations for the authors):**
Line 166: How was the serum-IgG purified? (e.g., protein A, protein G).

Protein A purification was used, as now mentioned in the manuscript. Purified Igs were thus predominantly IgG1, IgG2 and IgG4, as indicated.

(2) Line 196: When mentioning affinities, it is preferable to present in molar units.

To facilitate comparisons, Ab concentrations were presented in µg/mL as in Simonis et al.

(3) Line 206: The author states that P3D6 displays significantly reduced ExoS-Bla injection (Figure 2B), but according to the presented table, ExoS-Bla inhibition was higher for P5B3. Additionally, when using "significantly", what was the statistical test that was used to evaluate the significance? Please clarify.

We thank the reviewer for pointing out this inconsistency. Indeed, the names of P3D6 and P5B3 were exchanged when building the table related to Figure 2B. The corrected version of this figure is now presented in the new version of the manuscript. An ANOVA was performed to evaluate the significance of the observed difference (adjusted p-values < 0.001) and it is now mentioned in the figure caption.

(4) Line 215: "P3B3" typo.

This was corrected.

(5) Figure 3B: Could the author explain the higher level of ExoS-Bla injection when using VRCO1 antibody compared to no antibody.

A slightly higher level of the median is observed in the case of three variants out of five. However, this difference is not statistically significant (p-value > 0.05).

(6) Supplement Figure 1: the presented grey area is not clear (is it the 95%CI?) and how was the IC50 calculated? With what model was it projected? Are the values for IC50 beyond the 100µg/mL mark a projection? It seems that projecting such greater values (such as the IC50 of over 400µg/mL for variant 5) is prone to high error probability.

The grey area represents the 95% confidence interval (95% CI) and it is now mentioned in the figure caption. The IC50 and 95% CI were both inferred by the dose-response drc R package based on a three-parameters log-logistic model and it is now explained in the Materials & Methods section. The p-values for IC50 beyond the 100µg/mL were below 0.05 but we agree that such extrapolation should be considered with precaution (see below our response to comment number 7).

(7) Line 227: The author describes that P5B3 has similar IC50 values towards variants 1-4, but the IC50 towards variant 5 is substantially higher with 400µg/mL, albeit the only difference between variant 4 and 5 is the switch position 225 Arg -> Lys which are very similar in their properties. Please provide an explanation.

As explained in our response to comment number 6, we agree that the comparison of IC50 that are estimated to be close or higher than the highest experimental concentration is somehow speculative. Indeed, we performed further statistical analysis that showed no significant difference between the IC50 toward the five PcrV variants of mAb P5B3. In contrast, the difference between the IC50 of mAbs P5B3 and P3D6 toward variant 1 is statistically significant. This is now explained in the manuscript.

(8) Line 233: Pore assembly: It is not clear how the data was normalized. The authors mention the methods normalization against the wildtype strain in the absence of antibodies, but did not elaborate clearly if the mutant strain has the same base cytotoxicity as the wild type. It would be helpful to show the level of cytotoxicity of the wild type compared to the mutant in the absence of antibodies to understand the baseline of cytotoxicity of both strains.

In these experiments we did not use the wild-type strain. As explained, the only strain that allows the measurement of pore formation by translocators PopB/PopD is the one lacking all effectors. All the experiments were done with this strain, and all the measurements were normalized accordingly.

(9) Figure 4: The explanation is redundant as it is clearly stated in the results. It would be better for the caption to describe the figure and leave interpretation to the results section. Overall, this comment is relevant to all figure captions, as it will reduce redundancy. My suggestion is to keep the figure caption as a road map to understand what is shown in the figure. For example, the Figure 4 caption should include that the concentration is presented in logarithmic scale, what is the dashed line, what is the grey area (what interval does it represent?), what each circle represents, and what is the regression model used?

Figure captions have been improved as suggested.

(10) Line 432: The authors apparently misquoted the original article describing the chimeric form PcrV* by describing the fusion of amino acids 1-17 and 136-249. I quote the original article by Tabor et al. "[...] we generated a truncated PcrV fragment (PcrVfrag) comprising PcrV amino acids 1-17 fused to amino acids 149-236 [...]". Additionally, how does the absence of amino acid 21 in the variant affect the conclusion?

Our construct was inspired by the one described in Tabor et al. but was not identical. We have therefore replaced "was constructed based on a construct by Tabor et al." for "whose design was inspired by the construct described in Tabor et al."

Amino acid 21 is only absent in the construct used for crystallization experiments; all other experiments looking at Ab activity were performed with bacteria bearing full-length PcrV. The difference in P3D6 activity between variants V1 and V2-appears to be explained by the nature of the residue at position 225, according to the structural data, as explained now in more detail in the manuscript. Accordingly, the difference in efficiency of P3D6 against the V1 and V2 variants is explained by the residue at position 225, as both variants have the same residue at position 21. However, while the nature of the residue at position 225 appears to explain the absence of efficiency of the Ab for the variants studied, an impact of residue 21 could not be totally ruled out in putative variants with a Ser at 225 but different amino acids at 21.

(11) Line 569: Missing word - ESRF stands for European Synchrotron Radiation Facility.

This has been corrected.

(12) Line 268-269 (Figure 5A): The description of the alpha helices in relation to the figure is incomplete. Helices 2,3 and 5 are not indicated.

Indeed, since the structure is well-known and in the interest of visibility and simplicity, we only included the most relevant secondary structure features.

(13) Line 271-272: It would be good to elaborate on the exact binding platform between LC and HC of the Fab and the residues on the PcrV side. For example, the author could apply the structure to PDBePISA (EMBL-EBI) which will provide details about the interface between the PcrV and the antibody. It is very interesting to learn what regions of the antibody are in charge of the binding, such as: is the H-CDR3 the major contributor of the binding or are other CDRs more involved? Additionally, in line 275 they state that the substitution of Ser 225 with Arg or Lys is consistent with the P3D6 insufficient binding. What contributed to this result on the antibodies side?

In order to address this question, we are now providing a LigPlot figure (supplementary Figure 3) in which specific interactions between PcrV* and the Fab are shown.

(14) Line 291: It is unclear from what data the authors concluded that anti-PscF targets 3 distinct regions of PscF.

The data are shown in Supplementary Table 2, as mentioned in the manuscript. We have now modified the order of the anti-PcrV mAbs in the table to better illustrate the three identified epitope clusters (Sup table 2). Similarly, the anti-PscF mAbs appear to group into three clusters as P3G9 and P5E10 only compete with themselves, while mabs P3D6 and P5B3 compete with themselves and each other.

(15) Line 315: It is preferable to introduce results in the results section instead of the discussion.

While preparing the manuscript, we initially included these results as a separate paragraph in the Results section, but ultimately chose the current format to improve flow and avoid redundancy.

(16) Supplement Figure 2: What was the regression model used to evaluate IC50, and what is presented in the graph? What is the dashed line (see comment for Figure 4 above)?

The regression is based on a three-parameters log-logistic model and the light-colors area correspond to the 95% IC. The dashed lines visually represents 100% of ExoS-Bla injection. These information are now mentioned in the figure caption.

(17) Figure 6B: It would be better to show an additional rotation of the PcrV bound by Fab 30-B8 that corresponds to the same as the one represented with Fab MEDI3092. This would clear up the differences in binding regions. Same for Fab P3D6.

Figure 6 already depicts two orientations. Despite the fact that we agree that additional orientations could be of interest, we believe that this would add unnecessary complexity to the figure, and would prefer to maintain the figure as is, if possible.

(18) Line 356-358: The author proposes an experiment to support the suggested mechanism of P3D6, it would follow up with a bio-chemical analysis showing the prevention of PcrV oligomerization in its presence.

We understand the reviewers’ comment regarding the potential use of biochemical approaches to test our hypothesis. However, this not currently feasible as we have been unable to achieve in vitro oligomerization of PcrV alone, possibly due to the absence of other T3SS components, such as the polymerized PscF needle.

(19) Line 456: Missing details about how the ELISA was conducted including temperature, how the antigen was absorbed, plate type, etc.

Experimental details have been added.

(20) Line 460: Missing substrate used for alkaline phosphatase.

The nature of the substrate was added to the methods.